

# Time series of the summertime diurnal variability in the atmospheric water vapour isotopic composition at Concordia station, East Antarctica

Inès Ollivier[1,2] [*], Thomas Lauwers[2] [*], Niels Dutrievoz[2], Cécile Agosta[2], Mathieu Casado[2], Elise Fourré[2],
Christophe Genthon[3], Olivier Jossoud[2], Frédéric Prié[2], Hans Christian Steen-Larsen[1], Amaëlle Landais[2]

[1]Geophysical Institute, University of Bergen, and Bjerknes Centre for Climate Research, Bergen, Norway
[2]Laboratoire des Sciences du Climat et de l'Environnement, LSCE/IPSL, CEA-CNRS-UVSQ, Université Paris-Saclay, Gif sur Yvette, France
[3]Laboratoire de Météorologie Dynamique, LMD/IPSL, Sorbonne Université-CNRS, France
[*] These authors contributed equally to this work

*Correspondence to*: Inès Ollivier and Thomas Lauwers (ines.ollivier@uib.no, thomas.lauwers@lsce.ipsl.fr)

**Abstract.** Measurements of stable water isotopes in the atmospheric water vapour can be used to better understand the physical processes of the atmospheric water cycle. In polar regions, it is a key parameter to understand the link between the precipitation and snow isotopic compositions and interpret isotope climate records from ice cores. In this study we present a
novel 2.5-month record of the atmospheric water vapour isotopic composition during the austral summer 2023-2024 at Concordia Station (East Antarctica), from two independently calibrated laser spectrometers (CRDS and OF-CEAS measurement techniques) which are optimised to measure in low humidity environments. We show that both instruments accurately measure the summertime diurnal variability in the water vapour $\delta^{18}$O, $\delta$D, and d-excess when the water vapour mixing ratio is higher than 200 ppmv. We compare these measurements to the outputs of the isotope-enabled atmospheric
general circulation model LMDZ6-iso and show that the model exhibits biases in both the mean water vapour isotopic composition and the amplitude of the diurnal cycle, consistent with previous studies. Hence, this study provides a novel dataset of the atmospheric water vapour isotopic composition on the Antarctic Plateau, which can be used to evaluate isotope-enabled atmospheric general circulation models. The dataset is available on the public repository Zenodo (https://doi.org/10.5281/zenodo.14569655, Landais et al., 2024b).

## 25   1 Introduction

Stable water isotopes are unique tools to study the atmospheric water cycle, as they integrate information along successive phase changes. The relative abundances of the most common isotope species are expressed as $\delta^{18}$O and $\delta$D values, in per mill (‰) (Craig, 1961). The second order parameter deuterium excess (d-excess = $\delta$D - 8·$\delta^{18}$O, Dansgaard, 1964), has been defined to capture kinetic fractionation during phase changes throughout the hydrological cycle.



In polar ice cores, $\delta^{18}O$ and $\delta D$ have been traditionally interpreted as a temperature proxy based on empirical relationships between the mean annual temperature and the isotopic composition of snow samples (e.g. Johnsen et al., 1992; Jouzel et al., 2007; Lorius et al., 1979). Alongside, d-excess has been interpreted as a proxy for climatic conditions at the evaporative source region (e.g. Landais et al., 2021; Stenni et al., 2010; Uemura et al., 2008; Vimeux et al., 1999). However, an increasing number of studies have shown that this relationship between $\delta^{18}O$, $\delta D$, d-excess and climatic conditions is

affected by post-depositional processes at the ice sheet's surface (e.g. Casado et al., 2018, 2021; Ollivier et al., 2024; Steen-Larsen et al., 2014; Town et al., 2024; Zuhr et al., 2023). Specifically, the atmospheric water vapor isotopic composition above the ice sheet plays an important role on the isotopic signal found in the snow and firn through water vapor exchange during sublimation and condensation cycles (Dietrich et al., 2023; Hughes et al., 2021; Madsen et al., 2019; Ritter et al., 2016; Wahl et al., 2021, 2022).

Measurements of the atmospheric water vapour isotopic composition therefore provide key information on the processes at play at the ice sheet's surface and the link between water isotope records in the snow and firn and climatic conditions. In addition, such measurements can be used to evaluate the performances of isotope-enabled Atmospheric General Circulation Models (isoAGCMs hereinafter) (e.g. Risi et al., 2010; Werner et al., 2011, Dutrievoz et al., *in review*) beyond the common evaluation with surface snow samples that have been affected by post-depositional processes.

However, measuring the isotopic composition of water vapour in low humidity conditions below 500 ppmv, such as encountered on the East Antarctic Plateau, is a technical challenge since most laser spectrometers are designed for measuring accurately within a range of humidities between 5,000 and 30,000 ppmv. The vapour $\delta^{18}O$ and $\delta D$ measured by laser spectrometers strongly depends on humidity levels, which has to be taken into account for the calibration of the instruments (Casado et al., 2016; Landais et al., 2024a; Leroy-Dos Santos et al., 2021; Steen-Larsen et al., 2013). This can lead to

corrections larger than the amplitude of the diurnal signal (Leroy-Dos Santos et al., 2021).

At Concordia Station, on the East Antarctic Plateau, previous measurements of the water vapour isotopic composition have been limited in time (few weeks in December and early January, Casado et al., 2016, Leroy-Dos Santos et al., 2021) and associated with uncertainties as large as 5 and 20‰ for $\delta^{18}O$ and $\delta D$, respectively, when the humidity was below 200 ppmv. Therefore, there is a need to have measurements of the water vapour isotopic composition that are more accurate and over

longer time periods.

In this study, we present a time series of $\delta^{18}O$, $\delta D$ and d-excess of the atmospheric water vapour at Concordia Station, with an improved analytical precision compared to previous measurements. We installed two new laser spectrometers adapted for low humidity measurements, together with a calibration unit also designed to generate low humidity levels (Leroy-Dos Santos et al., 2021). The two analysers are based on different measurement techniques (Cavity Ring Down Spectroscopy -

CRDS - and Optical Feedback Cavity Enhanced Absorption Spectroscopy - OF-CEAS), which permits to compare both instrumental techniques in the low humidity conditions at Dome C and evaluate the performance of the OF-CEAS instrument, which has never been successfully measuring in the field at such low humidities. The thorough calibration of both instruments permitted the production of a coherent and accurate 2.5-month long time series of the water vapour isotopic



composition at Concordia Station over the austral summer 2023-2024. We further use this novel dataset to compare with
outputs from the isoAGCM LMDZ6-iso, as an example on how the dataset can be used for model evaluation.

## 2 Methods and data

### 2.1 Instrumental set-up

Concordia station is located on the East Antarctic plateau in the vicinity of Dome C (75.10° S, 123.33° E) at an altitude of
3233 m above sea level and about 1000 km away from the coast. The site is characterised by a mean annual temperature of -
52°C (Genthon et al., 2021).

The instrumental set-up for the continuous analysis of the water vapour isotopic composition (Fig. 1) presented in this study
is installed in an underground "shelter", a heated facility (+10°C) located 800 m upwind from the main station buildings
(75.10°S, 123.30°E). The setup is composed of (i) a heated sampling line, (ii) two laser spectrometers based on different
techniques optimised for water vapour isotope analysis at low humidities and (iii) a homemade low humidity generator to
perform automatic calibrations (LHLG, Leroy Dos Santos et al., 2021). The sampling line is a 16-meter long 1/4"
perfluoroalkoxy (PFA) line with an inlet situated about 50 cm above the snow surface (Fig. 1a). The line is insulated and
equipped with a heating cord to ensure a positive temperature and prevent condensation of water vapour. Water vapour is
pumped through the line with a typical flow of 10 L min$^{-1}$ and sent into the heated underground shelter, where the
calibrations and the measurements with both analysers are performed (Fig. 1b).


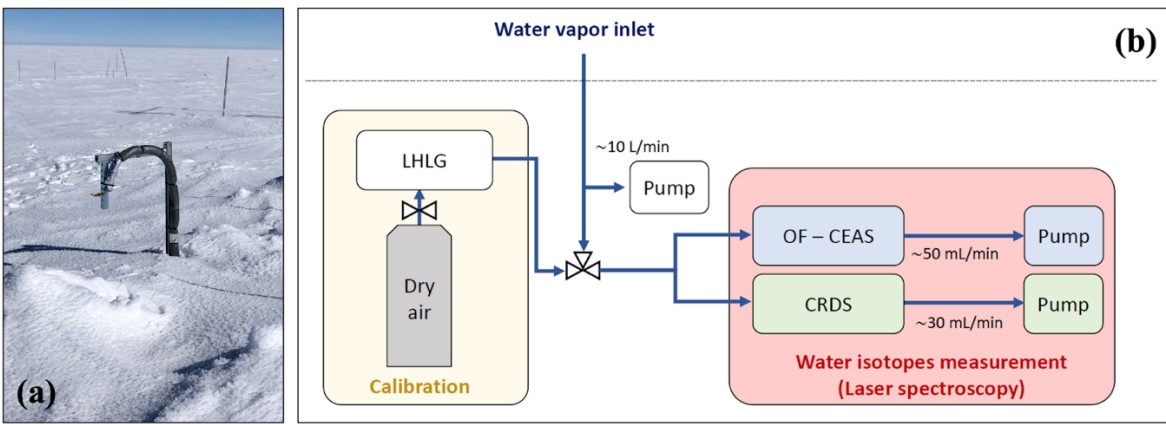

**Figure 1: Schematic of the instrumental set-up for the continuous analysis of the water vapour isotopic composition at Dome C.
Panel (a) shows a picture of the sampling line inlet above the snow surface. Panel (b) shows a schematic of the instrumental set-up
with both analysers and the calibration unit (LHLG) inside the heated underground shelter.**

The atmospheric water vapour isotopic composition is measured continuously in parallel by two distinct laser spectrometers,
respectively based on the CRDS technique and the OF-CEAS technique. The CRDS technique is based on an indirect
measurement of molecular absorption through the photon lifetime measurement inside a highly reflective resonant cavity.

The OF-CEAS measurement technique also relies on an optical cavity to increase the signal to noise ratio but directly measures the transmitted light. In addition, this technique uses optical feedback to stabilise the laser emission frequency, 90 enabling a lower instantaneous noise compared to the CRDS technique.

A CRDS analyser, manufactured by Picarro company (Picarro analyser hereinafter), was first installed in the summer season 2014-2015 for a test season and a new analyser was installed permanently in 2018 at Concordia station (Picarro HIDS2319). These instruments, coupled to the calibration unit, have proven to be robust and adapted for field measurements (Casado et al., 2016; Leroy-Dos Santos et al., 2021). However, increasing uncertainties on the signal below 300 ppmv led to restrict the 95 studies to December and January at Concordia station. Due to instrumental issues, the Picarro analyser HIDS2319 was replaced during the summer season 2021-2022 by a new analyser (Picarro HIDS2308 hereafter). The data presented in this study were collected by the latter. In parallel to the Picarro analyser, an OF-CEAS analyser manufactured by AP2E company (AP2E analyser hereinafter, Lauwers et al., 2024) was installed during the summer season 2022-2023 and optimised during the summer season 2023-2024. In this study we focus on the most recent austral summer period 2023-2024 (December to 100 mid-March), where both AP2E and Picarro analysers have been optimised and measuring in parallel on site.

## 2.2 Calibration protocols

In order to produce accurate atmospheric water vapour content and isotope measurements, we perform a series of calibration steps on the data provided by the two laser spectrometers. The mixing ratios measured by both instruments are calibrated against independent humidity measurements (Sect. 2.2.1). The raw isotopic ratios are corrected for the isotope-humidity 105 dependence of both analysers (Sect. 2.2.2) and then calibrated against the VSMOW-SLAP scale (Sect. 2.2.3). Lastly, section 2.2.4 presents the uncertainty estimation of the final calibrated measurements.

### 2.2.1 Calibration of the water vapour mixing ratio

To evaluate the accuracy of the measurement and calibrate the humidity measured by both analysers, we compare it to an independent in-situ measurement of the atmospheric humidity between January and March 2024. Note that data from this 110 independent measurement was not available in December 2023, so the comparison is restricted to the beginning of 2024 although the analysers were operating in December 2023.

The independent humidity sensor is installed about two meters above the surface and about twenty meters away from the inlet of the laser spectrometers. The sensor is an adapted HMP155 sensor, specifically designed to accurately measure the atmospheric humidity in dry and cold environments with frequent supersaturation conditions (Genthon et al., 2017, 2022). 115 As in Genthon et al. (2017), Vignon et al. (2022) and Ollivier et al. (2024), we use the data from the adapted HMP155 to recalculate the relative humidity with respect to ice. The relative humidity with respect to ice is then converted to water vapour mixing ratio (in ppmv) using the equations from Murphy and Koop (2005) together with the air pressure given by ERA5. Note that the resulting water vapour mixing ratio is not sensitive to the possible mismatch between the pressure given



by ERA5 and the local atmospheric pressure (not shown). We use this independent humidity measurement as the true
atmospheric humidity content to correct the humidity measured by the Picarro and AP2E analysers, as follows:

$$hum_{corr} = hum_{meas} \cdot slope_{hum} + int_{hum} \tag{1}$$

Where $hum_{meas}$ is the raw humidity given by the analyser (either AP2E or Picarro), $hum_{corr}$ is the humidity corrected on the
independent measurement and the coefficients $slope_{hum}$ and $int_{hum}$ are determined by a linear regression between the $hum_{meas}$
and the independent humidity measurement. The results of the linear regressions are presented in Sect. 3.1.1.

**2.2.2 Influence of humidity on the measured isotopic ratios**

For continuous water vapour isotopic measurement, and in particular in the East Antarctic plateau where mixing ratios are
often below 500 ppmv, both OF-CEAS and CRDS techniques are affected by the dependency of isotopic measurements on
the water vapour mixing ratio (e.g. Lauwers et al., 2024; Weng et al., 2020). We refer to this effect as the humidity-isotope
response. This humidity-isotope response is instrument-specific (e.g. Steen-Larsen et al., 2013) and is dependent on the
isotopic composition of the laboratory standard used to perform the calibrations (e.g. Lauwers et al., 2024; Weng et al.,
2020). A calibration of this dependency is therefore required in the humidity range of the site and using laboratory standards
with a known isotopic composition close to what is observed on site.
We determined the humidity-isotope response curves by performing one series of calibrations in January 2024. The
calibration curves for both analysers are determined using a single custom laboratory standard (FP5, $\delta^{18}O$ = -50.52 ± 0.05‰
and $\delta D$ = -394.7 ± 0.7‰), calibrated against the VSMOW-SLAP scale. We assume that the humidity-isotope response of
both analysers (AP2E and Picarro) is stable in the range of isotopic values measured on site, which was validated for a
Picarro analyser in Leroy Dos Santos et al. (2021). The standard FP5 has an isotopic composition closest to the water vapour
isotopic composition measured on site (from -50 to -80‰ in $\delta^{18}O$ and from -550 to -400‰ in $\delta D$ during summer months and
it has been previously used to calibrate a Picarro laser spectrometer at the same site (Leroy-Dos Santos et al., 2021). The
calibration steps were performed from high to low humidity (humidities ranging from 1100 to 50 ppmv). The humidity
levels are generated using the newest version of the custom calibration unit (LHLG, Leroy-Dos Santos et al., 2021), which
enables the generation of a steady water vapour flux with a known and stable isotopic composition.
The reference humidity for the calibration curves is set to 500 ppmv (see also Sect. 2.2.3). The results of the different
calibration steps are fitted with inverse functions (in combination to a linear function), as done in previous studies (e.g.
Lauwers et al., 2024). The coefficients of the inverse fits are used to correct the raw isotope data for the humidity-isotope
response, as follows:

$$\delta_{i,humcorr} = \delta_{i,meas} - \left[\frac{1}{c_1} \cdot hum_{meas} + c_2 \cdot hum_{meas} + c_3\right] \tag{2}$$



Where $\delta_{i,meas}$ is the raw isotope data given by the instruments (subscript i is for any isotope species, $\delta^{18}O$ or $\delta D$), $\delta_{i,humcorr}$ is the isotope data corrected for the humidity-isotope response of the instruments and the coefficients $c_1$, $c_2$, and $c_3$ correspond to the coefficients of the inverse functions fitted to the data of the calibration steps. Equation 2 is determined for each isotope species and each analyser. The results of the calibration steps, the inverse fits and the coefficients are presented in Sect. 3.1.2.

### 2.2.3 Absolute calibration of the measured isotopic ratios

In a second step, we perform the absolute calibration of both analysers to convert the raw isotopic compositions measured by the instruments (and corrected for humidity dependence beforehand) to isotopic values calibrated against the VSMOW-SLAP scale. Regular and automatic calibrations of both analysers are performed with two laboratory standards calibrated against VSMOW-SLAP (FP5: $\delta^{18}O$ = -50.52‰ and $\delta D$ = -394.7‰; NEEM: $\delta^{18}O$ = -33.5‰ and $\delta D$ = -257.2‰). The calibrations are performed every 48 to 72 hours with the LHLG, injecting both standards at a target humidity level of 500 ppmv. We use the isotopic ratios measured by both analysers during the calibrations between January 11th and June 6th 2024 to establish the linear equations for the absolute calibration of each instrument. To remove the influence of the humidity measured during each calibration on the measured isotopic ratios during the calibration step, we correct the isotopic ratios for the humidity-isotope dependence (Sect. 2.2.2). In addition, we discard the calibrations with a humidity outside of two standard deviations around the mean humidity and outside of two standard deviations around the mean isotopic ratio of all calibrations during the period. Because we do not observe any significant drift in the calibration data, we then average, for each laboratory standard and each analyser, the measured water isotopic composition of all the selected calibrations over the period and establish the linear equations against the true value of the standards. The linear functions for each analyser are used to calibrate the measurements against the VSMOW-SLAP scale, as follows:

$$\delta_{i,VSMOW-SLAP} = \delta_{i,humcorr} \cdot slope_{VSMOW-SLAP} + int_{VSMOW-SLAP} \tag{3}$$

Where $\delta_{i,humcorr}$ is the isotope data corrected for humidity-isotope response (subscript i is for any isotope species, $\delta^{18}O$ or $\delta D$, see Sect. 2.2.2 and Eq. 2), $\delta_{i,VSMOW-SLAP}$ is the final corrected and calibrated against VSMOW-SLAP isotope data and the coefficients $slope_{VSMOW-SLAP}$ and $int_{VSMOW-SLAP}$ are determined by the linear regression between the measured and true values of the two laboratory standards. The results of the absolute calibration step are presented in Sect. 3.1.3. The corrected and calibrated time series of the water vapor isotopic composition from both analysers are presented in Sect. 3.2.

It should be noted that the two laboratory standards used to perform the absolute calibration both have an isotopic composition above the one usually measured on site in the atmosphere. We therefore assume that the linear relationships between the true and measured $\delta^{18}O$ and $\delta D$ values can be extrapolated beyond the isotopic composition of both standards to be able to calibrate the in-situ measurements. Such assumption was validated for a Picarro analyser in Casado et al. (2016).



### 2.2.4 Estimation of measurement uncertainty

We present two approaches to estimate the uncertainty on the water vapour isotopic measurements. First, we propagate the uncertainty related to the measurement noise driven by low humidity measurements and day-to-day instrumental drift, which is manifested in the regular measurements of the two laboratory standards. Secondly, we carry out a Monte-Carlo simulation propagating the uncertainty of the absolute calibration against VSMOW-SLAP into the uncertainty estimate on the final calibrated water vapour isotope measurements.

We consider two sources of uncertainty associated with the $\delta^{18}O$ and $\delta D$ measurements. The first source of uncertainty follows a power law with respect to humidity due to the increase in measurement noise at lower humidity levels for both Picarro and AP2E analysers (Lauwers et al., 2024). The second uncertainty originates from the instrumental instabilities at hourly to daily time scales caused by the sensitivity of the optical signal of laser spectrometers to several environmental factors, such as temperature or mechanical perturbations. We refer to the latter uncertainty as the "drift" uncertainty. We

group the two uncertainties (noise at low humidity and drift) into the following formulation to estimate the combined uncertainty on $\delta^{18}O$ and $\delta D$ measurements:

$$\sigma_i(h) = \left(\sigma_{i,drift} \cdot h_{ref}\right) / h \qquad (4)$$

With $h_{ref}$ is the reference humidity of the calibration steps ($h_{ref}$ = 500 ppmv, Sect. 2.2.2 and 2.2.3) and h is the humidity measured by the laser spectrometers. $\sigma_{i,drift}$ corresponds to one standard deviation of the measured isotopic ratios (subscript i is for any isotope species, $\delta^{18}O$ or $\delta D$) of all the calibration steps performed over six months with two laboratory standards (selected calibrations steps, see Sect. 2.2.3).

The uncertainty is calculated for the whole dataset for both analysers and is valid from 50 to 1100 ppmv (i.e. corresponding

to the upper and lower limit of the humidity-response curves, see Sect. 2.2.2). With this method, the uncertainty on the data incorporates both the instrumental drift over six months, similarly as done by Casado et al. (2016), and the dependency of the uncertainty on the measured humidity (i.e. larger uncertainties at lower humidities). This measurement uncertainty is probably overestimated, as $\sigma_{i,drift}$ integrates both the drift from the LHLG and from each isotope analyser over a six-month period. The uncertainty $\sigma(h)$ for d-excess is calculated by propagating the uncertainties on $\delta^{18}O$ and $\delta D$, as follows:


$$\sigma_{d-excess}(h) = \sqrt{\sigma_{\delta D}(h)^2 + \sigma_{\delta^{18}O}(h)^2} \qquad (5)$$

Alternatively, we propose to compute the uncertainty on the final $\delta^{18}O$ and $\delta D$ values from the Picarro and AP2E analysers by performing a Monte Carlo test with 1000 resamples of the linear regression coefficients within their uncertainty range to

calibrate the $\delta^{18}O$ and $\delta D$ values against the VSMOW-SLAP scale (as described in Sect. 2.2.3 but including uncertainty on the linear equation coefficients in Eq. 3). The uncertainty (referred as $\sigma_{MC}$) is computed as one standard deviation of the 1000





Monte Carlo calibrated time series and should be similar to $\sigma_{drift}$, since the same dataset of calibration steps is used for both methods. We compute the uncertainty for d-excess by propagating the uncertainties on $\delta^{18}O$ and $\delta D$, using Eq. 5. Results of this analysis are presented in Sect. 3.1.4.

## 2.3 Model description

The LMDZ-iso model (Laboratoire de Météorologie Dynamique Zoom model equipped with water isotopes, Risi et al., 2010) is the isotopic version of the atmospheric general circulation model LMDZ6 (Hourdin et al., 2020). The version of LMDZ used here is nearly identical to the one used for the phase 6 of the Coupled Model Intercomparison Project (CMIP6, Eyring et al., 2016). The LMDZ6 model employs the Van Leer moisture advection scheme for the passive transport of water isotopes (Risi et al., 2010; Van Leer, 1977). The equilibrium fractionation coefficients between water vapour and liquid or ice phases are derived from Merlivat and Nief (1967) and Majoube (1971a, 1971b). The non-equilibrium (kinetic) fractionation coefficients are formulated by Merlivat and Jouzel (1979) for evaporation from the sea surface and by Jouzel and Merlivat (1984) for snow formation at supersaturation. We performed a simulation with the standard Low Resolution (LR) grid of LMDZ6 with a horizontal resolution of 2.0° in longitude and 1.67° in latitude (144×142 longitude-latitude grid). The simulation has 79 vertical levels, and the first atmospheric level is located around 10 m above ground level. The LMDZ-iso 3D-fields of temperature and wind are nudged toward the 6-hourly ERA5 reanalysis data with a relaxation time of 3 hours except below the sigma-pressure level corresponding to 850 hPa above sea level, where nudging is not applied. Surface ocean boundary conditions are taken from the monthly mean sea surface temperature and sea-ice concentration fields from the ERA5 reanalysis. The simulation is performed with a supersaturation parameter of 0.004 K$^{-1}$. The simulation covers the period from December 2023 to April 2024, with a 1-hourly resolution.

## 3 Results

### 3.1 Dataset calibration

### 3.1.1 Water vapour mixing ratio

Figure 2 shows the evaluation of the atmospheric mixing ratio (or humidity, in ppmv) measured by the two analysers (Picarro and AP2E) against an independent humidity sensor (Sect. 2.2.1). The humidity measured by both analysers agree very well with the independent humidity measurement, with linear regression slopes close to the one-to-one line for both analysers (Fig. 2a and b). Overall, the Picarro analyser measures a lower humidity content than the independent sensor (average difference of 20 ppmv between January 1[st] and March 15[th] 2024), especially at higher humidity levels (Fig. 2a). On the other hand, the AP2E analyser gives similar humidities than the independent sensor (average difference of 2 ppmv between January 1[st] and March 15[th] 2024) in the whole range of humidities (Fig. 2b).





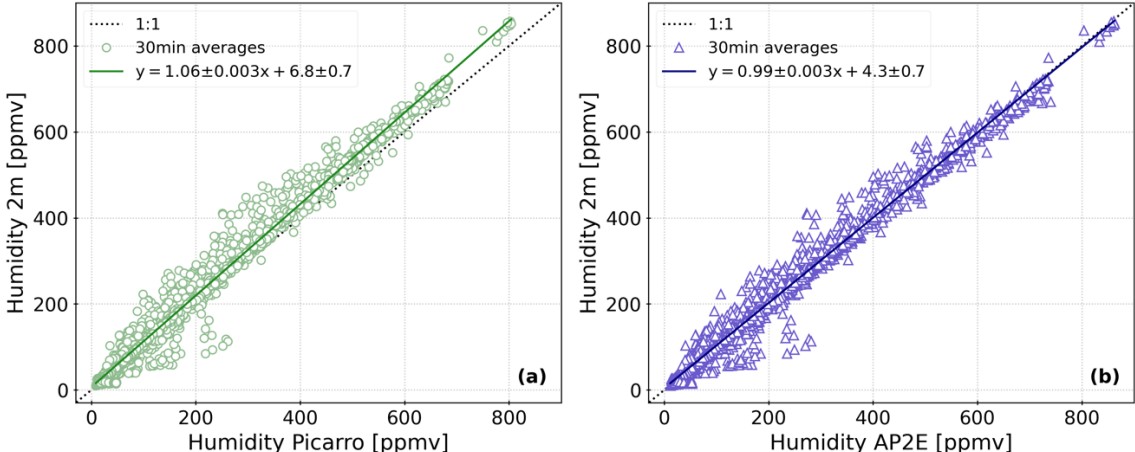

**Figure 2: Humidity (ppmv) measured by the two laser spectrometers (Picarro in panel (a) and AP2E in panel (b)) versus humidity measured by the independent sensor (modified HMP155, Sect. 2.2.1). All available 30 min averages between January 1st and March 15th 2024 are shown in the figure.**

Even if the difference between the humidity measured by the Picarro and AP2E analysers and the independent humidity sensor is small, the linear regression coefficients slope$_{hum}$ and intercepts at origin int$_{hum}$ (Fig. 2, Table 1) can be used to calibrate the humidity measured by both analysers, as described in Section 2.2.1.

**Table 1. Linear regression coefficients (shown in Fig. 2 and used in Eq. 1) for the correction of the humidity measured by both the Picarro and AP2E analysers.**

|  | slope$_{hum}$ [ppmv ppmv$^{-1}$] | int$_{hum}$ [ppmv] |
|---|---|---|
| Picarro HIDS2308 | 1.06 | 6.8 |
| AP2E | 0.99 | 4.3 |

During the period of interest (December 2023 to March 15th 2024), the humidity measured and calibrated by the two laser spectrometers ranges from 15 to 1100 ppmv (see also Fig. 6, Sect. 3.2). Note that the lowest humidity measured by the modified HMP155 system during this period is about 1 ppmv, however the two laser spectrometers didn't record this low humidity due to gaps in the dataset (Sect. 3.2).

**3.1.2 Humidity-isotope response**

Figure 3 shows the humidity-isotope calibration curves determined with the laboratory standard FP5 ($\delta^{18}$O = -50.52‰ and $\delta$D = -394.7‰, Sect. 2.2.2), for three laser spectrometers (described in Sect. 2.1): (1) the Picarro HIDS2319 analyser from Leroy-Dos Santos et al. (2021), (2) the Picarro HIDS2308 analyser (this study) and (3) the AP2E analyser (this study). For the Picarro HIDS2319 analyser, the calibration steps were performed with the initial version of the LHLG while for this study (Picarro HIDS2308 and AP2E analysers), the calibration steps were performed with the newest version of the LHLG

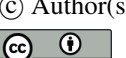



(Sect. 2.2.2). Each point on the humidity-response curves of all three analysers corresponds to the average isotopic composition of the calibration step over a ten-minute stable period. Note that each calibration step lasted from 40 min to 1h.

In Leroy-Dos Santos et al. (2021), the humidity-isotope response curves (for both $\delta^{18}$O and $\delta$D) of the Picarro HIDS2319 are described with polynomial fits (their equations 4 and 5, light green dashed lines in Fig. 3a and b). Their results show a

divergence of the measured isotopic composition below 500 ppmv, especially strong for $\delta$D (light green dashed line and dots in Fig. 3b). For the Picarro analyser HIDS2308, the humidity-isotope response curves are described with inverse fits (Sect. 2.2.2, dark green dotted lines in Fig. 3a and b). In comparison to the HIDS2319 analyser, the response curves show a similar strong divergence in $\delta^{18}$O and a much weaker divergence in $\delta$D. In addition, the HIDS2308 curves don't show any humidity-isotope dependence above 500 ppmv for both $\delta^{18}$O and $\delta$D (dark green dotted lines and dots in Fig. 3a and b). The difference

in humidity-isotope response of the two Picarro analysers (HIDS2319 and HIDS2308) is not surprising since different spectrometers will have a different humidity-isotope response (e.g. Steen-Larsen et al., 2013).

For the AP2E analyser, the humidity-isotope response curves are also described with inverse fits (Sect. 2.2.2, blue dotted lines in Fig. 3a and b). As already identified and described in Lauwers et al. (2024), the AP2E analyser humidity-isotope response curves show two different regimes. Below 500 ppmv, both $\delta^{18}$O and $\delta$D show a divergence with decreasing

humidity levels, in the opposite direction as for both Picarro analysers (blue dotted lines in Fig. 3a and b). Above 500 ppmv, $\delta^{18}$O shows a positive linear dependency to increasing humidity (blue dotted line in Fig. 3a), while a weaker dependency is observed for $\delta$D (blue dotted line in Fig. 3b).

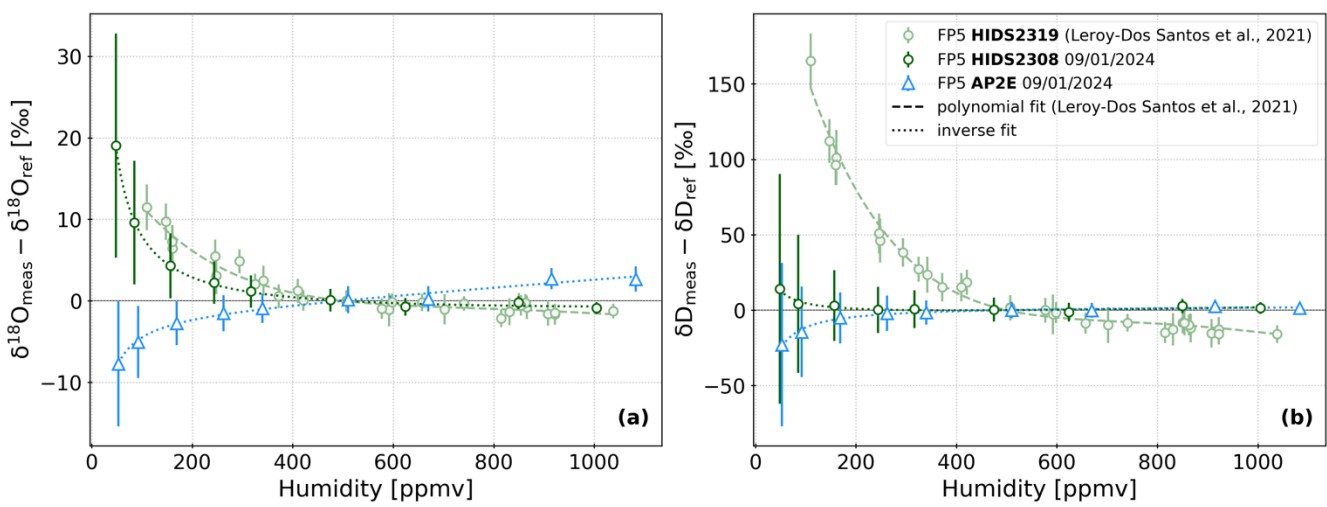

**Figure 3: Humidity-isotope ($\delta^{18}$O in panel (a) and $\delta$D in panel (b)) response curves for both the Picarro HIDS2319 (Leroy-Dos Santos et al., 2021), and the two laser spectrometers used in this study (Picarro HIDS2308 and AP2E analysers). The calibration steps and the data fitting are described in Sect. 2.2.2. In both panels, the dashed and dotted lines represent respectively a polynomial and inverse fit on the data. The error bars show the standard deviation around the 10-min average period of each calibration step of the three analysers (1σ, i.e. representation of the measurement noise). Note that to have the same reference**
**humidity (500 ppmv) for all three calibrations curves, the curves for the Picarro HIDS2319 were shifted downwards by the isotopic values of the polynomial fit at 500 ppmv (reference initially measured at 2000 ppmv, Leroy-Dos Santos et al., 2021).**





These results show that the isotope-humidity response of the Picarro analyser presented in this study is better constrained compared to the previous Picarro analyser, with a calibration curve determined down to a lower humidity than in Leroy-Dos Santos (50 ppmv in this study, 110 ppmv previously). In addition, the new Picarro shows a weaker isotope-humidity

dependence in the range of observed humidities at Dome C over the period of interest (15 to 1100 ppmv, Sect. 3.1.1), which leads to a better constrain on the correction for the isotope-humidity response and improves the reliability of the dataset. These results also show a well constrained isotope-humidity dependence for the AP2E analyser in the range of observed humidities at Dome C over the period of interest, which similarly to the Picarro analyser, improves the reliability of the dataset.

It should still be noted that the isotope-humidity calibration only goes down to 50 ppmv, although the minimum humidity recorded by the instruments is 15 ppmv during the period of interest (and the overall minimum humidity recorded by the HMP155 is 1 ppmv, Sect. 3.1.1). To correct the dataset, we therefore extrapolate the calibration curve down to 15 ppmv. This can lead to abnormal isotopic values after correction, leading to the increase of the uncertainty on the data at low humidities. This point is further developed in Sect. 3.1.4.

Table 2 summarises the coefficients of the inverse fits shown in Fig. 3 for both the Picarro HIDS2308 and AP2E analysers. As described in Sect. 2.2.2, these coefficients are used in Eq. 2 to calibrate the isotope measurements from both analysers for the humidity-isotope dependence (following Eq. 2, positive values in Fig. 3 correspond to a negative correction).

**Table 2. Coefficients of inverse functions (shown in Fig. 3 and used in Eq. 2) to calibrate the instruments for the humidity-isotope**
**response of both the Picarro and AP2E analysers.**

|  | $\delta^{18}O$ | | | $\delta D$ | | |
|---|---|---|---|---|---|---|
|  | $C_1$ [ppmv] | $C_2$ [ppmv$^{-1}$] | $C_3$ [ppmv ppmv$^{-1}$] | $C_1$ [ppmv] | $C_2$ [ppmv$^{-1}$] | $C_3$ [ppmv ppmv$^{-1}$] |
| Picarro HIDS2308 | 1024.9 | 0.0007 | -2.4 | 822.1 | 0.005 | -4.3 |
| AP2E | -336.2 | 0.005 | -1.6 | -1414.7 | 0.0005 | 2.6 |

### 3.1.3 Absolute calibration of isotopic ratios

As described in Sect. 2.2.3, the absolute calibration against the VSMOW-SLAP scale of the isotope data given by the Picarro and the AP2E analysers relies on the results of regular calibrations over six months of two laboratory standards with known

isotopic composition. Figure 4 shows the results of these regular calibrations performed between January and June 2024.

We first see that, despite a target humidity of 500 ppmv, the humidity measured during these regular calibrations varies slightly, from 250 to 450 ppmv, depending on which instrument and standard is measured (Fig. 4a). We also see that some of the calibrations are associated with very low humidities (red markers in Fig. 4a), which we exclude in the pool of calibrations used for the absolute calibration of both analysers (Sect. 2.2.3). These low humidity calibrations can be

explained by the LHLG, which failed to generate the target humidity level.

We observe that the measured $\delta^{18}O$ by both analysers varies throughout the period, but no drift is observed (Fig. 4b). Since the $\delta^{18}O$ values shown in Fig. 4b are corrected for the humidity-isotope response (Sect. 2.2.3), variations around the mean



$\delta^{18}O$ over the whole period cannot be explained by the variations of the humidity measured by the analysers (Fig. 4a). Instead, these variations can be explained by variations of environmental conditions, such as the temperature in the room

where the spectrometers are installed, or instability of the humidity generated by the LHLG during the calibration step. Despite these variations, the standard deviation of the ensemble of $\delta^{18}O$ values associated to the calibration of the two laboratory standards is low for both instruments (1.0‰ for the standard NEEM measured by the AP2E analyser and 0.8‰ for FP5; 0.8‰ for the standard NEEM measured by the Picarro analyser and 0.6‰ for FP5; Fig. 4b) compared to results from Lauwers et al. (2024) obtained at Dumont d'Urville station over a year. We further exclude the few calibrations which

appear as outliers (outside of two standard deviations around the mean $\delta^{18}O$, red markers in Fig. 4b) to establish the absolute calibration of both analysers (Sect. 2.2.3).

As for $\delta^{18}O$, we do not observe any drift in $\delta D$ over the period, for neither analyser (Fig. 4c). The standard deviation of the ensemble of $\delta D$ values associated with the calibration of the two laboratory standards is low for both instruments (7.4‰ for the standard NEEM measured by the AP2E analyser and 6.5‰ for FP5; 6.9‰ for the NEEM standard measured by the

Picarro analyser and 2.4‰ for FP5; Fig. 4c). These results are comparable with the results from Lauwers et al. (2024). For both laboratory standards, the variations in $\delta D$ over the period are higher for the AP2E analyser, which is probably due to the small absorption peak of $\delta D$ in the spectral window used by the analyser. We further exclude the few calibrations which appear as outliers (outside of two standard deviations around the mean $\delta D$, red markers in Fig. 4c) to establish the absolute calibration of both analysers (Sect. 2.2.3).


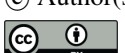

**Figure 4: Results of the regular calibrations performed with two laboratory standards (FP5 and NEEM) between January 11th and June 6th with the new version of the LHLG (description in Sect. 2.2.2 and 2.2.3). Panel (a) shows the humidity measured by both analysers during each calibration. The red markers show the calibrations that were discarded (outside of two standards deviations around the mean humidity). Panels (b) and (c) show the measured isotopic ratios by both analysers during each calibration as a deviation to the mean over the whole period. The isotopic ratios of each calibration are corrected for the isotope-humidity response of each analyser. In panels (b) and (c), only the accepted calibration from panel (a) are shown. The red markers show the calibrations that are discarded in a second step (outside of two standard deviations around the mean isotopic ratio).**

As described in Sect. 2.2.3, the results of the regular calibrations over six months are used to calibrate the data against the VSMOW-SLAP scale (selected calibrations from Fig. 4). Figure 5 shows the result of the linear regressions between the true and humidity-corrected $\delta^{18}O$ and $\delta D$. The coefficients of the linear regressions (used in Eq. 3) for both analysers and both isotope species are summarised in Table 3.





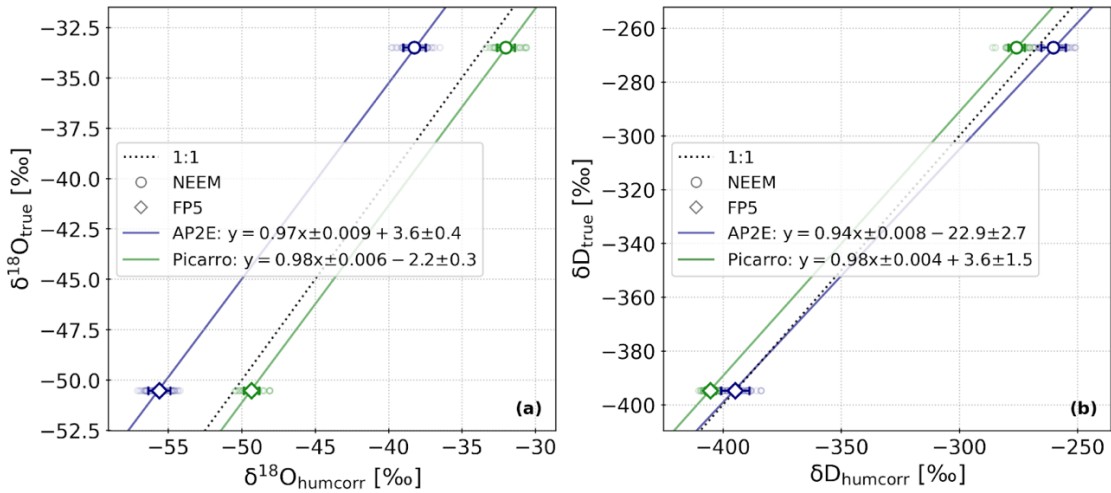

**Figure 5:** Humidity-isotope corrected ratios vs true isotopic ratios ($\delta^{18}$O in panel (a) and $\delta$D in panel (b)) of two laboratory standards (FP5 and NEEM) for both Picarro and AP2E analysers. In both panels, the smaller coloured markers represent all selected calibrations and the larger coloured markers the average isotopic ratio of all selected calibrations (whiskers represent one standard deviation). The coloured lines show the linear regressions between the true and humidity-corrected isotopic ratios using two laboratory standards.

**Table 3. Coefficients from the linear regressions between the true and humidity-corrected isotopic ratios using two laboratory standards (shown in Fig. 5 and used in Eq. 3) to calibrate the data from both Picarro and AP2E analysers against VSMOW-SLAP. The uncertainty associated with each coefficient corresponds to the standard error of the estimated coefficient (given by the *linregress* function from the python package *scipy*).**

| | $\delta^{18}$O | | $\delta$D | |
|---|---|---|---|---|
| | Slope$_{\text{VMOSW-SLAP}}$ [‰ ‰$^{-1}$] | Int$_{\text{VMOSW-SLAP}}$ [‰] | Slope$_{\text{VMOSW-SLAP}}$ [‰ ‰$^{-1}$] | Int$_{\text{VMOSW-SLAP}}$ [‰] |
| Picarro HIDS2308 | $0.98 \pm 0.006$ | $-2.2 \pm 0.3$ | $0.98 \pm 0.004$ | $3.6 \pm 1.5$ |
| AP2E | $0.97 \pm 0.009$ | $3.6 \pm 0.4$ | $0.94 \pm 0.008$ | $-22.9 \pm 2.7$ |

Both the Picarro and AP2E analysers have an absolute calibration slope for $\delta^{18}$O close to one (respectively 0.98 and 0.97, Fig. 5a and Table 3), which shows that both analysers capture the linearity between the true $\delta^{18}$O value of the two laboratory

375 standards. The intercepts of the linear relations for the two analysers are of the same magnitude, however opposite signs (-2.2‰ for Picarro and 3.6‰ for AP2E, Fig. 5a and Table 3). This indicates that the absolute calibration of the AP2E analyser will be of opposite sign and slightly larger than the Picarro, which is also visible in Fig. 6 and 7. The associated error on both linear coefficients from the two analysers are also comparable, despite the ones for the AP2E analyser being slightly higher (Fig. 5a and Table 3).

380 For $\delta$D, the Picarro shows an absolute calibration slope also close to one (0.98), while the AP2E analyser shows a lower slope (0.94, Fig. 5b and Table 3). This indicates that the Picarro also captures the linearity of the true $\delta$D value of the two laboratory standards, while the AP2E requires a stronger correction to calibrate the data against VSMOW-SLAP. Similarly, the intercepts of the linear relations are very different and of opposite signs between the two analysers (3.6‰ for Picarro and





-22.9‰ for AP2E, Fig. 5a and Table 3). This shows that the AP2E analyser is measuring further away than the true isotopic composition compared to the Picarro, and therefore that the correction to calibrate the AP2E analyser will be stronger than the one for the Picarro. This is also visible in Fig. 6 and 7. The results of the linear regressions for $\delta D$ also show that the errors associated to the coefficients for the AP2E analyser are twice as high than the ones for the Picarro (Fig. 5 and Table 3). This means that the error on the absolute calibration of the AP2E analyser is higher, as also described in the following section (Sect. 3.1.4).

### 3.1.4 Measurement uncertainty

In Eq. 4 (Sect. 2.2.4), $\sigma_{i,drift}$ is estimated as one standard deviation of the selected calibrations over six months, combining both laboratory standards (Fig. 4b and c). Table 4 summarises the values of $\sigma_{i,drift}$ found for $\delta^{18}O$ and $\delta D$ and for each analyser. Associated with the measured atmospheric humidity, this provides the measurement uncertainty on the final $\delta^{18}O$, $\delta D$ and d-excess from both analysers presented along the data in the following section.

**Table 4. Values of $\sigma_{i, drift}$ from Eq. 4 in Sect. 2.2.4 for both $\delta^{18}O$ and $\delta D$ and both laser spectrometers.**

|  | $\sigma_{drift}$ for $\delta^{18}O$ [‰] | $\sigma_{drift}$ for $\delta D$ [‰] |
|---|---|---|
| Picarro HIDS2308 | 0.6 | 2.9 |
| AP2E | 0.8 | 5.6 |

Besides, the Monte Carlo tests show that between December 2023 and January 2024, the uncertainty ($\sigma_{MC}$) of $\delta^{18}O$ from the Picarro is 0.5‰ and 2.7‰ for $\delta D$, which leads to an uncertainty of 3.0‰ on d-excess. The AP2E analyser shows higher uncertainties, with $\sigma_{MC} = 0.8$‰ for $\delta^{18}O$, 4.9‰ for $\delta D$, and 5.3‰ for d-excess. As expected, the errors $\sigma_{MC}$ on $\delta^{18}O$ and $\delta D$ are in the same order of magnitude as the corresponding $\sigma_{drift}$ (Table 4), since they are computed with the same set of calibrations (Sect. 2.2.4).

### 3.2 Time series of the water vapor isotopic composition

Figure 6 shows the evolution of the atmospheric humidity, $\delta^{18}O$, $\delta D$ and d-excess measured by both laser spectrometers between December 2023 and March 15th 2024. Figure 7 shows a focus on a four-day period in January 2024 (corresponding to the grey hatched area in Fig. 6). Note that the time series are not continuous, with interruptions due to calibration periods, maintenance work on the instruments or electrical shutdowns.

**Figure 6:** Time series (December 6th 2023 to March 15th 2024) of the atmospheric humidity (in ppmv, panel (a)), $\delta^{18}O$ (in ‰, panel (b)), $\delta D$ (in ‰, panel (c)) and d-excess (in ‰, panel (d)) measured by the Picarro (green lines) and AP2E (blue lines) analysers. In panels (b), (c) and (d), the green and blue shaded areas correspond respectively to σ(h) (Sect. 2.2.4) of the Picarro and AP2E analysers. In all four panels, the dashed lines correspond to the raw data given by the spectrometers and the plain lines correspond to the corrected and calibrated data (see Sect. 2.2 and 3.1). The grey hatched area marks the period from January 11th to January 15th shown in Fig. 7.


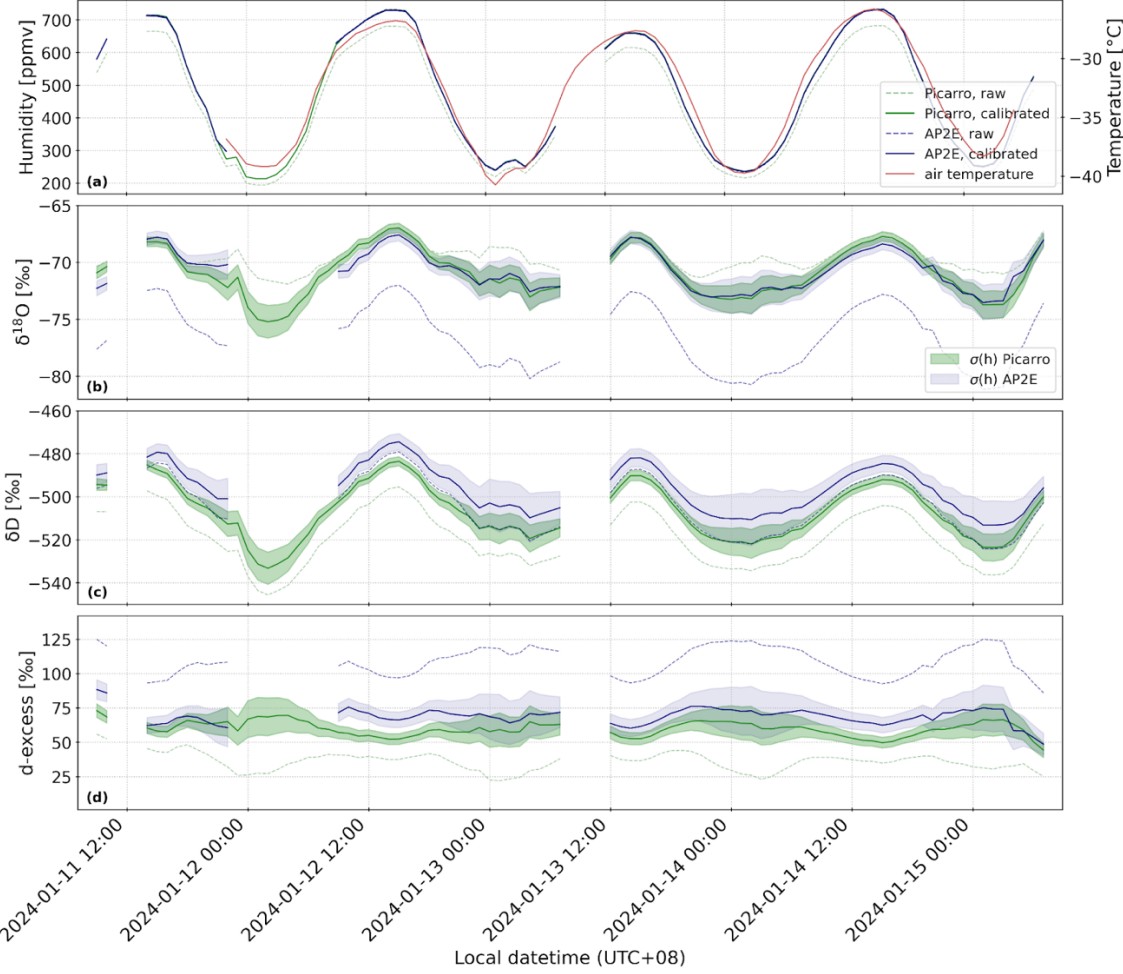

**Figure 7: Zoom on the period from January 11ᵗʰ to January 15ᵗʰ 2024 of the atmospheric humidity (in ppmv, panel (a)), $\delta^{18}O$ (in ‰, panel (b)), $\delta D$ (in ‰, panel (c)) and d-excess (in ‰, panel (d)) measured by the Picarro (green lines) and AP2E (blue lines) analysers. In panels (b), (c) and (d), the green and blue shaded areas correspond respectively to σ(h) (Sect. 2.2.4) of the Picarro and AP2E analysers. In all four panels, the dashed lines correspond to the raw data given by the spectrometers and the plain lines correspond to the corrected and calibrated data (see Sect. 2.2 and 3.1). In panel (a), the red line corresponds to the observed air temperature.**

The raw humidity measured by both analysers are in excellent agreement (Fig. 6 and 7), and the calibration against the independent humidity sensor has only a small effect. The calibrated humidities are showing the same diurnal variations for both analysers, synchronous with the temperature diurnal cycle on site (Fig. 7a). In addition, both instruments record the decrease of the humidity from the beginning of February, coinciding with the onset of the winter at Dome C (Fig. 6a).

Contrary to the humidity, the calibration of the raw data has a significant effect on the $\delta^{18}O$ time series of both analysers. For the AP2E analyser, the calibration of the raw $\delta^{18}O$ time series shifts it towards higher values (Fig. 6 and 7), with a mean difference of 6‰ over December and January between the raw and calibrated time series. This shift is expected from the



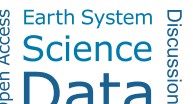

absolute calibration curve (Sect. 3.1.3). The amplitude of the diurnal cycle is also slightly reduced after applying the calibration (Fig. 7), due to the humidity-$\delta^{18}$O response of the analyser (i.e. positive correction for low humidities and negative correction for high humidities, Sect. 3.1.2). For the Picarro analyser, no systematic shift is observed before and after the calibration of $\delta^{18}$O (Fig. 6 and 7). However, the amplitude of the diurnal cycle is larger after calibration, as expected from the humidity-$\delta^{18}$O response of the Picarro which shows negative correction for lower humidities (Fig. 3, Sect. 3.1.2).

This is further visible on the period from the end of January onwards, where the diurnal cycles show an opposite behaviour between the raw and calibrated data: the raw data is in opposite phase to the humidity (minimum $\delta^{18}$O associated with maximum humidity) and the calibrated data is in phase with the humidity (minimum $\delta^{18}$O associated with minimum humidity). This is an effect of the large humidity-$\delta^{18}$O response of the Picarro at low humidities (Fig. 3, Sect. 3.1.2).

Compared to $\delta^{18}$O, the raw and calibrated $\delta$D time series from both instruments are rather similar, at least during the period

where the humidity is above 200 ppmv (mid-December to end of January, Fig. 6c). The calibration of both analysers slightly increases the average $\delta$D values (+7‰ for the AP2E analyser and +12‰ for the Picarro analyser on average over December and January). The calibration of the $\delta$D time series does not affect the amplitude of the diurnal cycle for neither analyser (Fig. 7c). Both raw $\delta$D time series compare relatively well from mid-December to the end of January (dashed lines in Fig. 6c), with the same in-phase relationship between $\delta$D and the mixing ratio as for the calibrated $\delta^{18}$O time series. This in-phase

relationship between $\delta$D and the humidity is preserved after calibration (plain lines in Fig. 6c and 7c).

There is a good agreement between the $\delta^{18}$O calibrated time series from the AP2E and Picarro analysers between mid-December to mid-February, confirming that the calibration is valid for the range of humidities encountered over this period. The mean difference in $\delta^{18}$O between the two instruments over the same period is 2‰, within the range of uncertainties of

the calibrated time series (Fig. 6). After mid-February, with humidity levels consistently below 200 ppmv (Fig. 6a), there is an increasing difference between the AP2E and Picarro analysers (Fig. 6).

As for $\delta^{18}$O, we observe that the calibrated $\delta$D time series from both instruments agree well between mid-December to mid-February (Fig. 6). There is a mean difference in $\delta$D between the two instruments of 8‰ over this period, which is also within the uncertainty of both calibrated time series (Fig. 6 and 7). Similarly to $\delta^{18}$O, we observe that after mid-February, the

calibrated $\delta$D time series of the two instruments start to diverge (Fig. 6).

Finally, the raw time series of d-excess are very different between the two analysers (Fig. 6 and 7). However, after the calibration of both analysers, the two d-excess time series are comparable within their uncertainty range (Fig. 6 and 7), with a mean difference between the two analysers of 7‰ between mid-December and mid-February. As for $\delta^{18}$O and $\delta$D, the calibrated d-excess time series of the two analysers diverge from mid-February onwards (plain lines in Fig. 6d).

The divergence in both $\delta^{18}$O and $\delta$D between the two instruments is probably due to the increase of instantaneous measurement noise of the analysers when the humidity decreases. It is also related to the difficulty of calibrating the instruments for very low humidity levels (Sect. 3.1.2). This is reflected in the uncertainty of the measurements (Fig. 6 and 7),





which increases significantly for both instruments from mid-February onwards, when the mixing ratio is consistently below

200 ppmv (Fig. 6). Because of these limitations, the comparison between the observations and the model presented in Sect.
3.3 is restricted to the period before mid-February.

In addition, the amplitude of the mean diurnal cycle in $\delta^{18}$O (calibrated data, calculated over the period January 11[th] to
January 15[th] 2024 shown in Fig. 7b) is similar for both instruments: 5.7‰ for the Picarro analyser (from -73.4 to -67.7‰, not
shown) and 4.7‰ for the AP2E analyser (from -72.6 to 67.9‰, not shown). The mean diurnal cycle in $\delta$D over the same
period is also comparable for both analysers: 34.9‰ for the Picarro analyser (from -523.2 to -488.3‰, not shown) and
29.5‰ for the AP2E analyser (from -509.6 to -480.1‰, not shown). As for $\delta^{18}$O and $\delta$D, both instruments show a similar
mean diurnal cycle in d-excess: 11.6‰ (from 52.9 to 64.5‰, not shown) for the Picarro analyser and 13.5‰ (from 63.2 to
76.7‰, not shown) for the AP2E analyser. Considering the uncertainties on the $\delta^{18}$O, $\delta$D and d-excess values of both
instruments, we conclude that both analysers compare well and that the AP2E captures well the diurnal cycle measured by
the Picarro analyser.

**3.3 Comparison of LMDZ6-iso outputs with novel in-situ measurements**

Recently, Dutrievoz et al. (*in review*) used in-situ observations of the water vapour isotopic composition at Concordia
Station to evaluate the performance of LMDZ6-iso to correctly capture the diurnal variations observed on site. This
comparison was performed over December 2018 and limited to $\delta^{18}$O due to the low confidence in the d-excess
measurements. Because of the large correction linked to the humidity-dependence on the $\delta^{18}$O signal, even the $\delta^{18}$O could be
challenged. We extend this comparison to the recent period December 2023 to mid-February 2024 using the novel and
reliable dataset presented in Sect. 3.2. Figure 8 shows the comparison of the humidity, $\delta^{18}$O, $\delta$D and d-excess over the whole
period. Figure 9 shows the same for a four-day period in January 2024 (corresponding to the grey hatched area in Fig. 8).

The comparison of the humidity modelled by LMDZ6-iso and measured by both analysers show an overall good agreement,
including for the amplitude of the observed diurnal cycle (Fig. 8a and 9a). However, during some specific periods, the model
shows higher humidity levels than what is observed, especially during the nighttime (e.g. December 16[th] to 20[th], light brown
area in Fig. 8a).
Although the model reproduces the observed in-phase relationship between $\delta^{18}$O and the humidity, the comparison between
the modelled and observed $\delta^{18}$O shows a poorer agreement than for humidity. Firstly, the modelled $\delta^{18}$O shows an overall
positive bias during the period December to mid-February compared to the observations, with a mean difference of 5.2‰
compared to the Picarro analyser and 3.3‰ compared to the AP2E analyser (Fig. 8b). Secondly, the amplitude of the diurnal
cycle modelled by LMDZ6-iso is overall larger than in the observations (Fig. 8b). Over the period January 11[th] to January
15[th] 2024 (Fig. 9b), the amplitude of the mean diurnal cycle in $\delta^{18}$O modelled by LMDZ6-iso is 10.9‰ (from -70.9 to -

60.0‰, not shown), higher than the one from both the Picarro analyser (5.7‰, Sect. 3.2) and the AP2E analyser (4.7‰, Sect.

3.2).

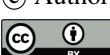

**Figure 8: Time series (December 5ᵗʰ 2023 to February 14ᵗʰ 2024) of the atmospheric humidity (in ppmv, panel (a)), $\delta^{18}$O (in ‰,**
**panel (b)), $\delta$D (in ‰, panel (c)), and d-excess (in ‰, panel (d)) measured (and calibrated) by the Picarro analyser (green lines), the**
**AP2E (blue lines) analysers, and modelled by LMDZ6-iso (grey lines). In panels (b), (c) and (d), the green and blue shaded areas**
**correspond respectively to σ(h) (Sect. 2.2.4) of the Picarro and AP2E analysers. In all four panels, the grey hatched area marks the**
**period from January 1ˢᵗ to 11ᵗʰ 2024 shown in Fig. 9 (same period as in Fig. 7). In panel (a), the light brown area marks the the**
**period from December 16ᵗʰ to 20ᵗʰ 2023 (period when the modelled and observed humidities differ).**

The same patterns are observed for $\delta$D. The modelled $\delta$D also shows an overall mean positive bias compared to the

observations, with a mean difference of 28.9‰ compared to the Picarro analyser and 20.9‰ compared to the AP2E analyser

(Fig. 8c). The amplitude of the diurnal cycle is also larger in LMDZ6-iso than in the observations (Fig. 8c). Between January



11th and January 15th 2024 (Fig. 9c), the mean diurnal amplitude modelled by LMDZ6-iso is 69.0‰ (from -515.8 to -446.8‰, not shown), which is higher than the observed one (34.9‰ for Picarro analyser, 29.5‰ for AP2E analyser, Sect. 3.2).

Lastly, due to the biases identified for $\delta^{18}$O and $\delta$D, the d-excess modelled by LMDZ6-iso also shows some discrepancies with the observations. The model shows an overall negative bias compared to the observations, with a mean difference between December and mid-February of 12.5‰ compared to the Picarro analyser and of 5.1‰ compared to the AP2E analyser (Fig. 8d). The comparison of the amplitudes of the diurnal cycle is less conclusive than for $\delta^{18}$O and $\delta$D, due to the large uncertainties associated with the observations (Fig. 9d). However, we observe that the model still correctly captures the observed anti-phase relationship between d-excess and $\delta^{18}$O (or $\delta$D), with a maximum d-excess when $\delta^{18}$O is minimal, i.e. during the night, and a minimum d-excess when $\delta^{18}$O is maximal, i.e. during the day (Fig. 9d).

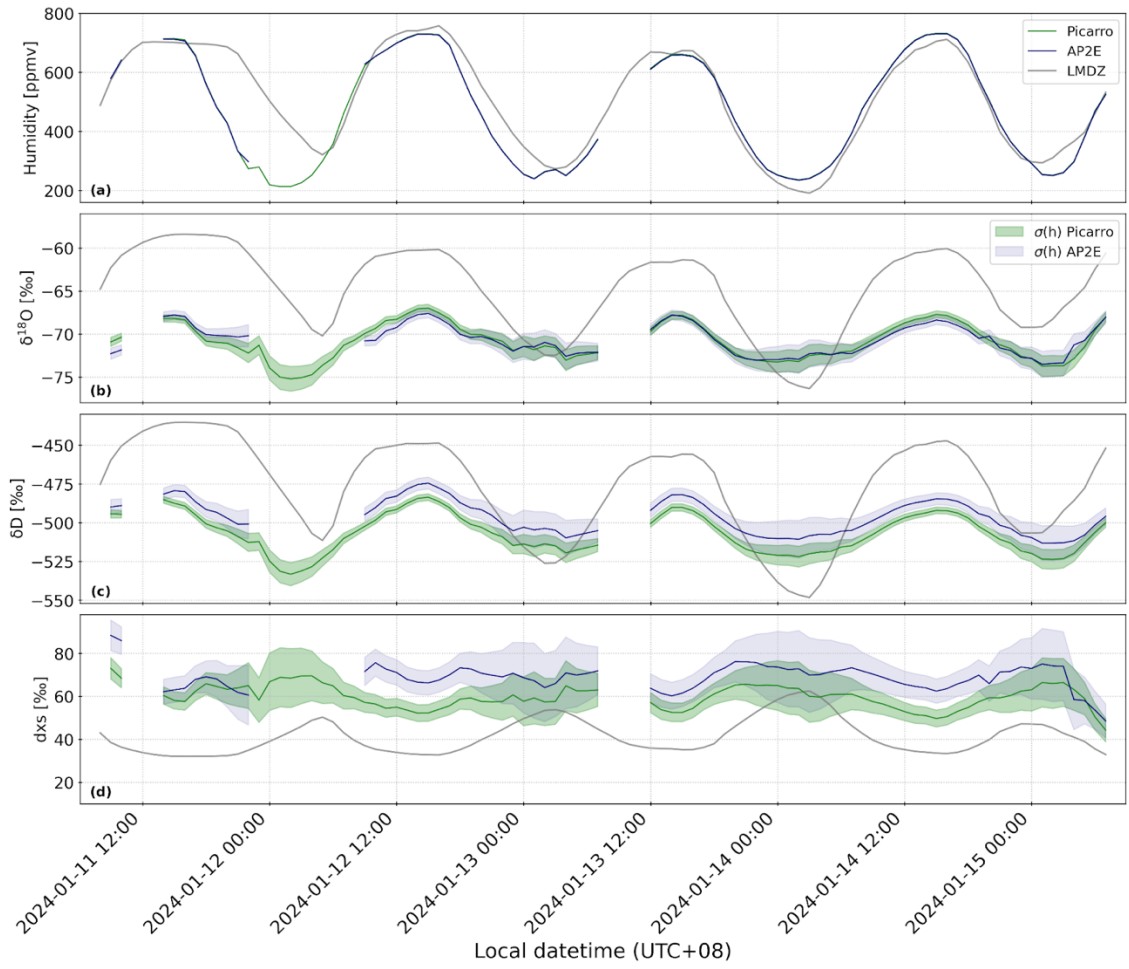

**Figure 9: Zoom on the period from January 11th to 15th 2024 of the atmospheric humidity (in ppmv, panel (a)), $\delta^{18}$O (in ‰, panel (b)), $\delta$D (in ‰, panel (c)) and d-excess (in ‰, panel (d)) measured by the Picarro analyser (green lines), the AP2E analyser (blue**





**lines), and modelled by LMDZ6-iso (grey lines). In panels (b) to (d), the green and blue shaded areas correspond respectively to σ(h) (Sect. 2.2.4) of the Picarro and AP2E analysers.**


Although the aim of this study is not to provide an in-depth evaluation of the LMDZ6-iso model, the discrepancies observed between the outputs of the model and the observations can provide indications on the possible biases in the model. This is discussed in the following section.

## 4 Discussion

We show that over the period from December $5^{th}$ 2023 to January $31^{st}$ 2024, there is a good agreement between the calibrated humidity, $\delta^{18}O$ and $\delta D$ time series from the AP2E and Picarro water vapour analysers. We therefore use this new dataset as the best measurements documenting the diurnal variability of water vapour isotopic composition during the summertime at Concordia Station. It permits to evaluate the humidity, $\delta^{18}O$ and $\delta D$ modelled by LMDZ6-iso for the lowest atmospheric level (0-7 m above the surface at Dome C). In general, there is a good agreement between the modelled and

observed humidity. The model also captures the observed evolution of the diurnal cycles of the water vapour isotopic composition. However, the model shows both a mean bias in the water vapour isotopic composition and a discrepancy in the amplitude of the daily cycle compared to the observations.

Our results support the conclusions from Dutrievoz et al. (*in review*), who showed larger amplitudes of the modelled $\delta^{18}O$

and $\delta D$ diurnal cycles in the model compared to observations. Dutrievoz et al. (*in review*) suggested that one explanation for this discrepancy could be that the model doesn't include the process of fractionation during sublimation, which has been shown to occur (Wahl et al., 2021). Sublimation generally enriches the snow surface in $\delta^{18}O$ and $\delta D$ (Casado et al., 2021; Hughes et al., 2021; Dietrich et al., 2023), which would lead to a decrease in the vapour $\delta^{18}O$ and $\delta D$ during the day (i.e. when sublimation occurs, coincides with higher humidity, $\delta^{18}O$ and $\delta D$ levels). Including fractionation during sublimation

could probably improve the comparison between the modelled and observed diurnal cycle of the water vapour isotopic composition. The discrepancy between the model and the observations could also arise from the ice-vapour equilibrium fractionation coefficients used in LMDZ6-iso (Sect. 2.3). These coefficients were established for temperatures down to -40 and -33°C, respectively, and extrapolated for lower temperatures. In addition, other fractionation coefficients from the literature disagree with the formulations from Merlivat and Nief (1967) and Majoube (1971a) (Ellehoj et al., 2013; Lamb et

al., 2017). Lastly, the amplitude of the water vapour isotopic composition diurnal cycle is also controlled by the amount of sublimation and turbulent mixing in the boundary layer during the day, and by condensation during the night. Although included in the model, these processes might not be well representing the in-situ conditions.

We also observe that mean values of both $\delta^{18}O$ and $\delta D$ in the water vapour are higher in LMDZ6-iso than in the

observations. On the other hand, the modelled vapour d-excess is, on average, lower than in the observations. The bias in the

modelled $\delta^{18}$O and $\delta$D was also identified by Dutrievoz et al. (*in review*), despite the high uncertainty associated with the measurements. This overall bias in the modelled vapour isotopic composition could be explained by the isotopic composition of the snow in LMDZ6-iso, which might differ significantly from the actual snow surface at Dome C. Indeed, the snow isotopic composition in LMDZ6-iso during December 2023 and January 2024 is higher (+1‰ in $\delta^{18}$O and +19‰ in $\delta$D) than
the observed mean isotopic composition of the snow surface in December and January (average over the period 2017-2021, Ollivier et al. 2024).

The water vapour isotopic measurements presented in this study provide important benchmarks to evaluate the performance of isoAGCMs. The discrepancies identified between LMDZ6-iso and the observations highlight issues in the model physics
and/or in the implementation of water isotopes in the model. Combining the observations of the water vapour isotopic composition with other meteorological observations brings new constraints to improve the representation of the Antarctic boundary layer in models and to reduce the uncertainty on isotopic fractionation coefficients at low temperatures. Both are needed to improve isoAGCMs in Antarctica, which in turn are needed for a better climatic interpretation of isotope records from Antarctic ice cores.

**5 Data availability**

Data described in this manuscript can be accessed at Zenodo under https://doi.org/10.5281/zenodo.14569655 (Landais et al., 2024b).

**6 Conclusions**

We have installed at Concordia Station two water vapour isotopic analysers using different optical spectroscopy techniques
and optimised for measuring at low humidities. The two instruments were carefully and independently calibrated with a dedicated calibration unit designed to generate low humidity levels. This permitted accurate measurements of the atmospheric water vapour isotopic composition at Concordia Station for a 2.5-month long period during the austral summer 2023-2024 and to validate the performance of the OF-CEAS measurement technique against CRDS for in-situ measurements. In addition, the thorough calibration of the instruments permitted to constrain the uncertainty on the datasets,
which can be used to evaluate isotope-enabled atmospheric general circulation models.
As a demonstration of the usefulness of the new dataset, we used this novel dataset to compare with the outputs from LMDZ6-iso, which shows two types of biases in the model outputs. The model first shows a mean bias of the water vapour isotopic composition over the study period (positive bias in $\delta^{18}$O and $\delta$D, negative bias in d-excess). In addition, the model


overestimates the amplitude of the diurnal cycle in the water vapour $\delta^{18}O$ and $\delta D$. This confirms the model-observations discrepancies identified by Dutrievoz et al. (*in review*).

The instruments installed at Concordia Station will continue to record the atmospheric water vapour isotopic composition in the upcoming years, to complement ongoing isotopic measurements of precipitation and snow (Dreossi et al., 2024; Ollivier et al., 2024) and to provide long-term measurements at this remote location on the East Antarctic Plateau. Further improvements are still needed to reduce the measurement uncertainties and to constrain the humidity-isotope calibration

curves down to very low humidities (below 100 ppmv) to be able to measure during the wintertime. This will be done by improving the accuracy of the calibration at very low humidity levels (e.g. by reducing the effect of residual water mixing effects) and through the development of a new generation of laser spectrometers (Casado et al., 2024).

*Author contribution.* IO, TL and AL designed the study and contributed to the analysis. IO and TL performed the data
curation and formal analysis, and IO performed the visualisation. IO, TL, MC, EF, OJ, FP and AL participated to the acquisition of the water vapor isotopic data. TL performed the field calibration and optimisation of the instruments. ND and CA performed the model simulations and provided inputs to the study. CG acquired the meteorological data and provided inputs to the study. HCSL provided inputs to the calibration protocol and data uncertainty estimation. IO, TL and AL prepared the manuscript draft, and all authors contributed to reviews and edits.


*Competing interests.* The authors declare that they have no conflict of interest.

*Acknowledgments.* This publication was generated in the frame of the DEEPICE and AWACA projects. The projects have received funding from the European Union's Horizon 2020 research and innovation programme under grant agreements no.
955750 (DEEPICE) and no. 951596 (AWACA). The water vapor isotopic data presented in this study has been collected within the frame of the French Polar Institute (IPEV) project NIVO 1110. We acknowledge using data from the project CALVA 1013 and GLACIOCLIM observatories supported by the French Polar Institute (IPEV) and the Observatoire des Sciences de l'Univers de Grenoble (OSUG) (https://web.lmd.jussieu.fr/~cgenthon/SiteCALVA/CalvaData.html, https://glacioclim.osug.fr/). We thank the logistics staff at Concordia Station and Manon Mastin for the instrumental
maintenance and help with the data acquisition.

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
