# Peer review of "Time series of the summertime atmospheric water vapour isotopic composition at Concordia station, East Antarctica"

_Earth System Science Data, 2025_

## Author Comment (AC1)

Response to anonymous referee #1

We thank Reviewer 1 for their time and effort to provide detailed and constructive feedback on the manuscript, which has improved the quality of the study. We have addressed all comments below and propose to implement the changes in a revised version of the manuscript.

Black: reviewer comment
Blue: author's response
Green: revised text

The article presents a temporal series of the isotopic composition of water vapor during the austral summer 2023/2024 at Dome C, East Antarctica. The data have been obtained using two different laser spectrometers: a Picarro Cavity Ring-Down Spectrometer (CRDS) and an AP2E Optical-Feedback Cavity Enhanced Absorption Spectrometer (OF-CEAS). According to Lauwers et al. (2024), the low-humidity OF-CEAS analyzer, which was supposedly deployed in the field for this paper, should perform better than the CRDS at low humidity, but in this study it seems to give worse results.

A comparison between the measured and the LMDZ6-iso simulated isotopic composition of water vapor has also been carried out in this paper. The modeled and measured data show a good agreement for humidity and also show an overall agreement for $\delta^{18}O$, $\delta D$ and, to a certain extent, for deuterium excess, but LMDZ6-iso, while capturing the general variability, fails to obtain the correct isotopic values as well as the magnitude of the observed diurnal cycles.

Line 15 the measurement period is more than 3 months rather than 2.5 months

We refer here to the part of the measurement period that is available in public access in Landais et al. (2024b), which is 2.5 months (December 6th 2023 – February 14th 2024). However, it is correct that the whole measurement period is longer than that (3.5 months). We chose to show the whole measurement period in the manuscript but explain why we only focus / make available 2.5 months of the dataset in the Section 3.2 of the manuscript. We also have improved the text regarding this matter (see our answer to the Reviewer 2 comment n°40).

Line 46-47 change "such as encountered on the East Antarctic Plateau, is a technical challenge since most laser spectrometers are designed for measuring accurately within a range of humidities between 5,000 and 30,000 ppmv" with "such as those encountered on the East Antarctic Plateau, presents a technical challenge, as most laser spectrometers are designed for measuring accurately within a range of humidities between 5,000 and 30,000 ppmv"

Modified.

Line 57-58 how is the commercially available Picarro laser spectrometer adapted for low humidity measurements?

The phrasing of our original sentence might be confusing. The commercial Picarro analyser is not adapted to low humidity measurements but calibrated for low humidities. It is only the AP2E analyser that was specifically designed for low humidities. To make this point clearer, we modified the text as follows (l. 57-59):

We installed a new laser spectrometer (ProCeas, AP2E Inc.) adapted for low humidity measurements (Lauwers et al., 2025) in parallel to a Picarro L2130-i laser spectrometer and together with a calibration unit designed to generate low humidity levels (Leroy-Dos Santos et al., 2021).

Line 63 same as line 15

See our answer to the reviewer's first comment.

Line 91-91 I think it would be better to specify the Picarro model (L2130-$i$? L2140-$i$?) instead of the identifier of your instrument. The same for the AP2E instrument: is it a ProCeas?

We use the identifier of our instruments to make the difference between the two different Picarro laser analysers that were installed at Concordia station (one of which the data is published in Leroy-Dos Santos et al., 2021). However we agree that the model of the analyser should be stated here. We modified the text as follows (l. 91-100):

A Picarro L2130-i analyser (Picarro Inc., CRDS measurement technique, Picarro analyser hereinafter) was first installed in the summer season 2014-2015 for a test season and permanently in 2018 at Concordia station (referred to as Picarro HIDS2319 hereafter). [...] Due to instrumental issues, the Picarro HIDS2319 was replaced during the summer season 2021-2022 by a new Picarro L2130-i analyser (referred to as Picarro HIDS2308 hereafter). [...] In parallel to the Picarro analyser, a prototype of a AP2E ProCeas analyser (AP2E Inc., OF-CEAS measurement technique, AP2E analyser hereinafter), adapted for low humidity measurements (Lauwers et al., 2025), was installed during the summer season 2022-2023 and [...]. In this study we focus on the austral summer period 2023-2024 (December to mid-March), where both Picarro and AP2E analysers have been measuring in parallel on site.

Line 139 change "closest" with "close"

Modified.

Line 140 Is it possible for the deuterium excess to be +90‰ (based on a $\delta^{18}O$ =-80‰ and a $\delta D$ =-550‰)?

The values for the atmospheric water vapour isotopic composition stated in this sentence correspond to published data from Leroy-Dos Santos et al. (2021). However, in their study they did not conclude on the atmospheric water vapour d-excess composition. In the new calibrated dataset presented in our study, we find a maximum d-excess of 93‰ ($\delta^{18}O$ = -77‰ and $\delta D$ = -523‰) during the period from December to mid-February. To make clear that the values given here are already published data, we slightly modified the text as follows (l. 139-141):

The standard FP5 has an isotopic composition close to the atmospheric water vapour isotopic composition measured on site (varying between approximately -50‰ and -80‰ in $\delta^{18}$O and between approximately -400‰ and -550‰ in $\delta$D during summertime, Leroy-Dos Santos et al., 2021) and it has been previously used to calibrate a Picarro [...].

Line 160-161 Although you give an explanation in line 180-183, I am still wondering why you used two laboratory standards mostly outside the range of the measured water vapor isotopic values. Couldn't you use, for example, the VSAEL standard with the FP5 standard?

We agree with the reviewer's comment on the fact that we used two standards mostly outside of the range of measured isotopic ratios. However, the standard mentioned here by the reviewer (VSAEL) was not yet made and available when the calibrations of the instruments were performed in the field. We therefore used the most depleted standard available to us at that time, which was the home-made FP5 standard.

Line 201-202 "i" is not subscript

Modified.

Line 273 What model is the Picarro HIDS2319?

Based on the reviewer's previous comment above, we added the information l. 91 about the model of the different Picarro analysers referred to in the text (they are all L2130-i analyzers).

Line 284-285 change "Below 500 ppmv, both $\delta^{18}$O and $\delta$D show a divergence with decreasing humidity levels, in the opposite direction as for both Picarro analysers" with "Below 500 ppmv, both $\delta^{18}$O and $\delta$D diverge as humidity levels decrease, but in the opposite direction observed in both Picarro analysers"

Modified.

Figure 4 $\delta^{18}$O$_{humcorr}$- $\delta^{18}$O$_{humcorr}$ [‰] and $\delta$D$_{humcorr}$- $\delta$D$_{humcorr}$ [‰] for the y axis is not very clear

We modified the vertical axes of Figure 4 to improve clarity. The axes are now: $\Delta\delta^{18}$O$_{humcorr}$ and $\Delta\delta$D$_{humcorr}$ (see our answer to Reviewer 2 comment n°34 for updated figure). We added the information of what it corresponds to in the figure caption as follows (l. 347-353):

Figure 4: Results of the regular [...]. Panels (b) and (c) show the measured isotopic ratios, corrected for the humidity-isotope response (Sect. 2.2.2 and 3.1.2), by both analysers during each calibration as a deviation of the mean over the whole period ($\Delta\delta^* = \delta^*{}_{humcorr} - \overline{\delta^*{}_{humcorr}}$, * is for any isotope). [...]

Line 374-375 and 381-382 what does it mean that "both analysers capture the linearity between the true $\delta^{18}$O value of the two laboratory standards"? A line passing through two point is always a linear equation

We agree with the reviewer's comment that these sentences are not very clear. Here we were highlighting that the slopes are close to 1, showing that the instruments don't need a large correction to calibrate the data against VSMOW-SLAP (except for the biases shown by the intercepts). We modified the text to incorporate this comment as follows:

l. 373-375:

Both the Picarro and AP2E analysers have an absolute calibration slope for $\delta^{18}O$ close to one (respectively 0.98 and 0.97, Fig. 5a and Table 3). The intercepts of the linear relations [...].

l. 380-382:

For $\delta D$, the Picarro [...]. This indicates that the AP2E analyser requires a stronger correction to calibrate [...].

Figure 7 I can't find "AP2E, raw" in panel a), which should be a blue dashed line; is it present in the graph? Was air temperature not available in the first and the last period of measurements? Where does the temperature come from? Is it AWS temperature or modeled temperature? You should specific it in the main text and in the figure caption

The "AP2E, raw" is present in the graph, superimposed by the "AP2E, calibrated" line, so not clearly visible. The temperature shown in red is the data from the 2-m level of a 42-m meteorological mast installed at Concordia station. However, as pointed out by the reviewer, there is some missing data at the beginning and end of the period and we do not explain where the data come from. We therefore used a different temperature dataset from the local AWS (Grigioni et al., 2022) to update the panel (a) of Figure 7. To accompany this change, we also modified the text and figure caption as follows:

l. 410:

[...] electric shutdowns. The air temperature shown in Fig. 7 is measured at 1.5 m above the surface by an Automatic Weather Station (AWS) installed in the vicinity of Concordia station (Grigioni et al., 2022).

l. 425:

In panel (a), the red line corresponds to the observed air temperature measured by the local AWS (Grigioni et al., 2022).

Line 455-456 there is a divergence between the two instruments between mid-February and mid-March. I know it's due to the very low humidity which makes it hard for the laser spectrometers to correctly measure the isotopic composition, but how do you explain the different behavior of the Picarro and the AP2E laser spectrometers and which one is more reliable? I think this is an important point if you wish to measure the isotopic composition of water vapor in other seasons

We attribute the different behaviour of the two analysers to the humidity levels. At these low humidities (below 50 ppmv, which is most of the time in March when the two instruments show a very different behaviour), the humidity-isotope response calibrations are not constrained for either analyser (lowest calibration point is 50 ppmv, as explained in Sect. 2.2.2). Therefore neither instrument is really reliable at these humidities because the calibration is extrapolated. This is also the reason why we discard this period for the observation-model comparison. However, it seems that this particular Picarro analyser would be best in capturing the atmospheric water vapour isotopic composition at these low humidities, as shown in Figure 6b and c where the diurnal cycle in $\delta^{18}$O and $\delta$D measured by the Picarro follows the diurnal cycle in the atmospheric humidity (minimum $\delta^{18}$O and $\delta$D associated to minimum in humidity), which is what we expect. Contrary to the Picarro, the AP2E shows an opposition of phase between $\delta^{18}$O and $\delta$D during March. Nevertheless, this does not permit concluding on the ability of the AP2E or Picarro analyser to measure at very low humidity levels, since the data shown here is only calibrated down to 50 ppmv. Further efforts should be put to constrain further the calibration in order to use the instruments at very low humidities.

Line 483 change "to correctly capture" with "in correctly capturing"

Modified.

Line 485-486 change "Because of the large correction linked to the humidity-dependence on the $\delta^{18}$O signal, even the $\delta^{18}$O could be challenged" with "Due to the significant correction associated with the humidity dependence of the $\delta^{18}$O signal, even the $\delta^{18}$O measurement could be questioned"

Modified.

Line 486-487 why you stopped the comparison at mid-February when you have data up to mid-March 2024? Is it because of the unreliability of the isotope data due to the very low humidity? I think it should be explained in the text

We decided to present the data until mid-March to show how the instruments agree at certain humidity levels and then start to diverge, and therefore only use the time period where both instruments are in good agreement to compare to the model. We explain this in the text l. 469-470. Further details are also given in our answer to Reviewer 2 comments n°5 and 40.

Line 490 change "show" with "shows"

Modified.

Line 491 change "including for the amplitude of the observed diurnal cycle" with "including in terms of the amplitude of the observed diurnal cycle"

Modified.

Line 495-496 change "the modelled $\delta^{18}$O shows an overall positive bias during the period December to mid-February compared to the observations" with "the modelled $\delta^{18}$O shows an overall positive bias during the entire period compared to the observations"

Modified.

Line 538 change "during the summertime" with "during summertime"

Modified.

Line 547-549 The higher deuterium excess in the measurements with respect to LMDZ6-iso could also be explained by sublimation

This is a very good point, and we added in the text a sentence to highlight this point as follows (l. 549-550):

[...] (coincides with higher humidity, $\delta^{18}$O and $\delta$D levels). Fractionation during sublimation would also affect the d-excess in the water vapour and could partly explain the discrepancy between the observed and modelled diurnal cycle in d-excess. Including fractionation during sublimation could therefore improve [...]

Line 556-557 change "might not be well representing the in-situ conditions" with "they may not accurately represent the in-situ conditions"

Modified.

Line 570-571 change "Combining the observations of the water vapour isotopic composition" with "Combining observations of water vapour isotopic composition"

Modified.

Lauwers, T., Fourré, E., Jossoud, O., Romanini, D., Prié, F., Nitti, G., Casado, M., Jaulin, K., Miltner, M., Farradèche, M., Masson-Delmotte, V., and Landais, A.: OF-CEAS laser spectroscopy to measure water isotopes in dry environments: example of application in Antarctica, https://doi.org/10.5194/egusphere-2024-2149, 15 August 2024.

Grigioni, P., Camporeale, G., Ciardini, V., De Silvestri, L., Iaccarino, A., Proposito, M., and Scarchilli, C.: Dati meteorologici della Stazione meteorologica CONCORDIA presso la Base CONCORDIA STATION (Dome C), ENEA [dataset], https://doi.org/10.12910/DATASET2022-002, 2022.

---

## Author Comment (AC2)

Response to anonymous referee #2

We thank Reviewer 2 for their time and effort to provide detailed and constructive feedback on the manuscript, which has improved the quality of the study. We have addressed all comments below and propose to implement the changes in a revised version of the manuscript.

Black: reviewer comment
Blue: author's response
Green: revised text

Review for Ollivier et al. (2025)

This manuscript documents a dataset of atmospheric water vapour isotope compositions observed during 6 Dec 2023 to 14 Feb 2024 at Concordia station over East Antarctica. The dataset represents a valuable effort of observations which require novel techniques to overcome harsh conditions at Dome C. The dataset is useful for evaluating isotope-enabled atmospheric models and investigating post-depositional modification of snow isotope compositions. The manuscript is well structured and clearly written. However, some improvements can be expected, as detailed below.

1, Line 1: Suggest removing "diurnal variability in the", as the data are not "time series of variability".

We modified the title of the manuscript with the following: "Time series of the summertime atmospheric water vapour isotopic composition at Concordia Station, East Antarctica".

2, Line 13: "it is a key parameter". What is a key parameter to "interpret isotope climate records from ice cores"? Isotope measurements from the ice cores? Please rephrase to be clearer.

We modified the text to make our point clearer (l. 13):

In polar regions, the atmospheric water vapour isotopic composition is a key parameter to [...]

3, Line 15: could you provide the exact start and end date of the record here?

We added the period covered by the recorded l. 14-15 as follows:

In this study we present a novel 2.5-month accurate record of the atmospheric water vapour isotopic composition during the austral summer 2023-2024 (December 6th 2023 to February 14th 2024) [...]

4, Line 16: Please provide the full name of "CRDS and OF-CEAS" before using the abbreviation.

For simplicity, we modified the text to prevent having to provide the full name of the measurement techniques in the abstract as follows (l. 16-17):

[...], from two laser spectrometers based on different measurement techniques, which are independently calibrated and both optimised to measure in low humidity environments.

5, Line 19: "higher than 200 ppmv". Based on Fig. 3, it seems the two laser spectrometers agree with each other up to lower limits in mixing ratios that are different for $\delta^{18}O$ and $\delta D$. Could you please provide quantitative estimates of the lower limits in mixing ratios that the two instruments agree? Is this agreement your criterion to decide whether the measurements are accurate? Could you also comment on the current valid humidity ranges for $\delta^{18}O$ and $\delta D$ measurements separately?

We have made the choice to state a validity range in time rather than in humidity range for this particular dataset. The reason why is because on the time series, we clearly identify a moment when the two datasets start to diverge mid-February, coinciding with the humidity dropping consistently below 200 ppmv. We therefore have established a "visual" threshold on humidity to determine which period should be used to compare to model outputs. As stated in our answer to the reviewer's comment n°40 (see below), when considering the data up to mid-February only (i.e. from when the humidity is consistently below 200 ppmv), the mean differences are reduced by half (for both $\delta^{18}O$ and $\delta D$) and the squared Pearson correlation coefficients are improved by 38% for $\delta^{18}O$ (from 0.58 to 0.8) and by 44% for $\delta D$ (from 0.59 to 0.85). This gives us confidence in the data over the 2.5 months period from December 6th to February 14th, which is the period we make available in public access and recommend to compare to model outputs.

We acknowledge that the 200 ppmv threshold chosen is a bit arbitrary and only valid for this specific dataset. However, for future studies, it would be very interesting to determine the humidity threshold for $\delta^{18}O$ and $\delta D$ separately to optimize the usability of the dataset, especially to isolate and study specific events such as atmospheric rivers. This was beyond the scope of our study, mainly because our goal was to provide all isotopes together over the same period (especially to have the opportunity to look at d-excess).

We have modified the sentence l. 19 to the following:
[...], when the water vapour mixing ratio is consistently higher than 200 ppmv.

Regarding Figure 3, we agree with the reviewer that the point at which the calibration curves start to diverge is in general associated with increasing calibration uncertainties, which are currently difficult to assess and were discussed in previous work (Lauwers et al., 2025). But we would like to emphasize that the low humidity divergence is a well documented spectroscopic artefact (e.g. Weng et al., 2020), and thus cannot be used as a criterion to decide whether the measurements are accurate. Instead, the most objective criterion we found was to use the inter-comparison between the instruments to evaluate the validity of the measurement

(through criteria such as MD and Pearson correlation coefficients, as suggested by the reviewer).

6, Line 19: "output" is uncountable.

Modified.

7, Line 21: "Hence…" Could you also mention the comparison of humidity, and then the added value of water isotopes for comparison? Based on the comparison, could you please briefly comment potential pathways to improve the modelled hydrological cycle?

As this manuscript is presenting the novel dataset, we decided to not delve too much on the comparison between the model and the observations. We submitted this manuscript to ESSD to focus on the dataset itself. However, we do provide a comparison with the LMDZ-iso model to show the usefulness of such a dataset, and provide potential pathways to improve the modelled hydrological cycle in the manuscript and refer the reviewer to the Discussion section (from l. 535). Dedicated work on the model biases and how it can be improved is currently done by the co-authors N. Dutrievoz and C. Agosta, such as published in Dutrievoz et al. (2025).

8, Line 29: The deuterium excess is not defined to capture kinetic fractionation, or at least it is not only affected by kinetic fractionation. Even equilibrium fractionation can result in a slope other than 8 and affect the value of deuterium excess. You can refer to this paper: Dütsch, M., Pfahl, S., & Sodemann, H. (2017). The impact of nonequilibrium and equilibrium fractionation on twodifferent deuterium excess definitions.Journal of Geophysical Research:Atmosphers, 122, 12,732–12,746 https://doi.org/10.1002/2017JD027085

This is a very good point and we have added this information to the text as follows (l. 28-29):

The second order parameter deuterium excess [...] hydrological cycle, although it can also be affected by equilibrium fractionation (e.g. Dütsch et al., 2017).

9, Line 35: As far as I understand, these studies show that snow isotopes can be modified post deposition, but how do these post-depositional processes affect the empirical relationships between isotopes and climate conditions? The empirical relationships using surface snow isotopes indeed already take the post-depositional processes into account, right?

Yes, this is a correct observation. We modified the text to make our point clearer as follows (l. 30-36, in bold):

In polar ice cores, $\delta^{18}O$ and $\delta D$ have been traditionally interpreted as a temperature proxy based on empirical relationships between the mean annual temperature and the isotopic composition of snow samples (e.g. Johnsen et al., 1992; Jouzel et al., 2007; Lorius et al., 1979). Alongside, d-excess has been interpreted as a proxy for climatic conditions at the evaporative source region (e.g. Landais et al., 2021; Stenni et al., 2010; Uemura et al., 2008; Vimeux et al., 1999).

However, an increasing **number of studies have shown that the isotopic composition (both $\delta^{18}$O, $\delta$D, and d-excess) of the snow surface deeper in the snowpack is affected by post-depositional processes at the ice sheet's surface** (e.g. Casado et al., 2018, 2021; Ollivier et al., 2025; Steen-Larsen et al., 2014; Town et al., 2024; Zuhr et al., 2023).

10, Line 53: What is the uncertainty when the humidity is above 200 ppmv? And how much is it reduced in this dataset?

The method used in Leroy-Dos Santos to estimate the uncertainty is different from the method we used in our manuscript, so it is a bit delicate to compare directly. Nevertheless, they estimate the uncertainty on $\delta^{18}$O to be approximately 2.5‰ when the humidity is maximal (between 400 and 600 ppmv, their Figure 7) and close to 4.5‰ when the humidity is minimal (around 200 ppmv, their Figure 7). In our study, we find the uncertainty on the $\delta^{18}$O measured by the Picarro to be between 0.3‰ at 1000 ppmv (approximate maximum value during studied time period) and 1.5‰ at 200 ppmv. Both values were calculated with Equation 4 and the values of $\sigma_{drift}$ in Table 4.

11, Line 60: "which permits" => "which permit". Could you please check through the manuscript to ensure no such mistakes? It can be easily done with tools like Grammarly.

Modified. And we have now checked the manuscript with Grammarly.

12, Line 62: Just curious whether it is because people never tried or they failed to do it successfully.

Such instruments have been deployed in the field before (including at Concordia station), but could not measure successfully. Major efforts in modifying the instruments themselves were made in recent years, and therefore could be re-deployed in the field.

13, Line 71: "The instrumental set-up … is installed"? Or just "The instrument"?

Here we refer to the overall instrumental set-up, comprising the two laser spectrometers, the sampling line, the calibration unit, etc. This overall set-up is installed in the heated shelter.

14, Line 72: "(+10°C)" => at a temperature of 10°C. Otherwise, it seems to be 10°C warmer than unheated ground.

Modified.

15, Line 75: 1/4", is this diameter or radius in what unit?

Here the number corresponds to the external diameter of the tube, in inches. We modified the text to include this information as follows (l. 75-76):

The sampling line is a 16-meter long perfluoroalkoxy (PFA) line (external diameter 1/4 in), with [...].

16, Line 87: Could you please provide reference/website/product description source to the CRDS and OF-CEAS techniques for interested readers? From Fig. 3 it seems that at low humidity levels, AP2E measurements are closer (further) to the reference for $\delta^{18}O$ ($\delta D$) than HIDS2308. Do the authors have an explanation for the differences based on technique attributes?

Following the Reviewer 1 comment n°5, we modified the text l. 91-100 as follows, where we also added the web links to the products as footnotes:

A Picarro L2130-i[1] analyser (Picarro Inc., CRDS measurement technique, Picarro analyser hereinafter) was first installed in the summer season 2014-2015 for a test season and permanently in 2018 at Concordia station (referred to as Picarro HIDS2319 hereafter). [...] Due to instrumental issues, the Picarro HIDS2319 was replaced during the summer season 2021-2022 by a new Picarro L2130-i analyser (referred to as Picarro HIDS2308 hereafter). [...] In parallel to the Picarro analyser, a prototype (non commercially available) of a AP2E ProCeas[2] analyser (AP2E Inc., OF-CEAS measurement technique, AP2E analyser hereinafter), adapted for low humidity measurements (Lauwers et al., 2025), was installed during the summer season 2022-2023 and [...]. In this study we focus on the austral summer period 2023-2024 (December to mid-March), where both Picarro and AP2E analysers have been measuring in parallel on site.

[1]https://www.picarro.com/environmental/products/l2130i_isotope_and_gas_concentration_analyzer, last accessed 30 July 2025
[2]https://www.ap2e.com/en/our-gas-analyzers/proceas/, last accessed 30 July 2025

Regarding the differences between the AP2E and Picarro analysers on Figure 3, they could partly be explained by the sensitivity of the measurement technique to humidity levels. However, they cannot be directly comparable as they are internal calibrations performed in the Picarro analyser that are not performed in the AP2E analyser (all is done post-processing).

17, Line 94: remove "led to"

Modified.

18, Line 115: Which quantity do you use to recalculate relative humidity relative to ice, and how? Do you need temperature information and where is it from?

The adapted HMP155 sensor provides three measurements simultaneously: the ambient temperature measured in a ventilated shield, and the temperature and relative humidity measured in a heated ventilated shield. By recombining these three measurements, the relative humidity with respect to ice can be calculated. We refer the reviewer to the Appendix A in Vignon et al. (2022) and Supplement S1 in Ollivier et al. (2025) where the calculations are detailed.

19, Line 121: Based on Fig. 2, it seems HMP155 measurements are larger than other two instruments on average, inferred based on the intercept. Could you please provide root mean squared error (RMSE), mean error (ME), and mean absolute error (MAE)? Do you think the systematic differences may result from different locations, heights, and inlets (one heated and how about the another one)? How much does these factors matter and how much does it matter if no humidity calibration is conducted? What is the difference between raw humidity measurements by the two instruments (RMSE, ME, MAE)?

We calculated the RMSE, ME (analyser – HMP155 or Picarro – AP2E) and MAE between the raw humidity provided by each analyser and the HMP155. For the Picarro analyser: RMSE = 32.3 ppmv, ME = -19.2 ppmv and MAE = 22.7 ppmv. For the AP2E analyser: RMSE = 23.0 ppmv, ME = -2.3 ppmv and MAE = 13.2 ppmv. The HMP155 and the analysers' inlet are in the very close vicinity (within 10 m radius), so the systematic differences can most probably be explained by difference in height (roughly 1.5 m difference) and / or uncertainty in the measurement provided by the HMP155 (possible problem with the heating system). By looking at the two raw humidity time series provided by the two analysers (Figures 6a and 7a), they agree well in capturing the diurnal variability, although the Picarro measures a lower humidity than the AP2E analyser (also visible in Figure 2, see quantification of difference below). Here we proceed to a humidity calibration to be able to compare to the model outputs, but this calibration does not affect any other results presented in the paper, as the calibrations performed for the isotopes is not dependent on the humidity calibration. So in our case, this calibration does not matter for the results we present for the isotopes. The difference between the raw humidity measurements from both analysers is summarized by a RMSE = 20.5 ppmv, ME = -15.9 ppmv and MAE = 15.9 ppmv. We modified Figure 2 to incorporate this information (see our answer to comment n°29 below).

20, Line 139: "closest" among what?

Closest among available home made laboratory standards. We modified the text as this was also commented by Reviewer 1. We modified "closest" to "close".

21, Line 142: How many calibrations did you conduct? Do you change the humidity level from 1100 to 50 ppmv for each calibration? Then how long does each humidity level last for the calibration and do you (need to) account for memory effects during consecutive humidity levels? In Fig. 3, it seems the calibration humidity is at some random levels and is different for two instruments, why?

We conducted 9 individual calibrations, each one of them associated with a single humidity (the humidity level does not vary within each calibration). Each calibration lasts for about two hours, and we average the last 10 minutes to get each data point. This way, we minimize the memory effect by assuring that the potential residual water from previous calibrations has been evacuated from the system and that the humidity level has reached a stable state. In addition, compared to measurement uncertainty, the memory effect is negligible at 200 ppmv. Nevertheless, memory effect is the reason why we do not go below 50 ppmv for the calibration steps, then the residual water is not negligible. In Figure 3, the calibration humidities are

different for both instruments because we show the raw humidity. As explained in our answer to comment n°19 above, we do not calibrate the isotopes with the calibrated humidity but with the measured humidity, to prevent incorporating an additional error. Therefore both instruments, although the target calibration humidity is the same, will not measure the same humidity and neither the precise target humidity (that is the reason why it also appears as random values).

We have added the information about the number of calibrations in the text as follows (l. 135):

[...] performing one series of nine calibrations in January 2024.

22, Line 145: what does a reference humidity mean?

The reference humidity is used to calibrate the data against a common reference humidity, which we choose to be 500 ppmv. This is common practice to calibrate the instruments for their humidity-isotope response (e.g. Steen-Larsen et al., 2013).

23, Line 176: "against VSMOW-SLAP"

Modified.

24, Line 202: 'any isotope species' => 'each isotope species'

Modified.

25, Line 211: Do you need to take account of the coefficient 8 while estimating uncertainties in dxs?

This is an oversight in Equation 5 in the manuscript, the factor 8 should indeed be included. This does not change the uncertainty on d-excess shown in the different figures of the manuscript, as it was calculated with the correct formulation (including the factor 8). We modified Equation 5 in the text as follows:

$$\sigma_{d-excess}(h) = \sqrt{\sigma_{\delta D}(h)^2 + 8 \times \sigma_{\delta^{18}O}(h)^2}$$

26, Line 216: "referred to"

Modified.

27, Line 221: provide the full name of LMDZ when you first mention it.

We moved the acronym description higher in the manuscript, l. 65:

[...] outputs from the isoAGCM LMDZ6-iso (isotope enabled version of the Laboratoire de Météorologie Dynamique Zoom model version 6), as an example [...]

And removed the acronym description l. 221:

The LMDZ-iso model is the isotopic version (Risi et al., 2010) of the atmospheric general circulation model LMDZ6 (Hourdin et al., 2020).

28, Line 235: how is the surface snow isotope composition over Antarctic ice sheet configured in LMDZ6-iso?

The isotopic composition of the snow over the Antarctic ice sheet is equivalent to a snow bucket which averages snowfall since the beginning of the simulation (Dutrievoz et al., 2025). We added this information in the text as follows (l. 234):

[...] fields from ERA5 reanalysis. In the model, the isotopic composition of the snow is equivalent to a snow bucket which averages snowfall since the beginning of the simulation (Dutrievoz et al., 2025). The simulation [...]

29, Line 248: As mentioned before, could you please provide RMSE, ME, and MAE between the independent analyzer and two instruments, as well as between the two instruments, ideally direct on the figure? There is an obvious cluster of outliers where two instruments indicate humidity larger than 200 ppmv and the independent humidity sensor gives values around 100 ppmv. Is this associated with large temperature deviations or different wind conditions? I would be interested to see a panel c where you plot Picarro vs. AP2E, which measure apple and apple. I am quite curious why the independent sensor is considered as ground-truth: is the technique more reliable, or are humidity sensors in Picarro and AP2E not optimized for humidity measurements, or humidity measurements may be biased by the heated sample line and the inlet?

We consider the independent sensor as ground-truth because of the measurement technique that is optimized for low humidity measurements such as encountered at Concordia Station (Genthon et al., 2017; Vignon et al., 2022). We did not find any dependence to wind speed or temperature deviations for the cluster of outliers. However, it is possible that due to the measurement uncertainty or the different measurement height between the HMP155 and the analysers, the comparison is not so straightforward. We rather use the data as an independent measurement to evaluate and calibrate the two laser spectrometers individually. Following comment n°19 above, we also compare the humidities measured by the two analysers and see that they agree very well, except for a systematic bias (see updated Figure 2 below). It is known that laser spectrometers have to be calibrated for the measurement of humidity, especially for Picarro. For the AP2E analyser, it is somehow already calibrated in the factory (that is also why the slope compared to the independent sensor is close to 1). To include this information in the manuscript, we updated Figure 2 as follows:

[Figure]

Figure 2: Comparison of the humidity (ppmv) measured by the two laser spectrometers (Picarro and AP2E) and by the independent sensor (modified HMP155, Sect. 2.2.1): (a) Picarro versus HMP155, (b) AP2E versus HMP155, and (c) Picarro versus AP2E. All available 30 min averages between January 1$^{st}$ and March 15$^{th}$ 2024 are shown in the figure. On each panel the root mean square error (RMSE, in ppmv), mean error (ME, analyser – HMP155 or Picarro – AP2E, in ppmv) and mean absolute error (MAE, in ppmv) calculated between the two humidity measurements are shown.

We also modified the text as follows (l. 245):

[...] humidities (Fig. 2b). The humidity measured by both analysers also compare very well together, with an overall positive bias of the AP2E compared to the Picarro (Fig. 2c).

30, Line 271: 'humidity-isotope response curves'.

Modified.

31, Line 278: "a much weaker divergence in $\delta$D". Could you briefly discuss what leads to such improvements? And Line 280, it is surprising to me to see such big differences.

The fact that different Picarro analysers might have a different humidity-isotope response is well known (Steen-Larsen et al., 2013; Weng et al., 2020). Since the humidity-isotope response is instrument dependent, it is not very surprising that in our study the two different Picarro analysers show a different response.

32: Line 290: Could you discuss why there is a tendency for Picarro to overestimate and AP2E to underestimate isotope ratios at low humidity levels? Could you quantitatively determine the applicable range of AP2E and Picarro for $\delta$D and $\delta^{18}$O separately? So it can serve as an objective baseline for comparisons in future works.

As mentioned in our answer to the previous comment, the humidity-isotope response is instrument dependent, such that each Picarro will give a different response (as shown on our Figure 3). It is possible that another Picarro analyser would give a similar response as the one seen for the AP2E analyser in our study (i.e., underestimate isotope ratios at low humidity levels), as for example shown in Steen-Larsen et al. (2013, their Figure S2). It is therefore not possible here to draw a general conclusion from our results, but rather explain the calibrations needed in the case of our study. For both $\delta^{18}$O and $\delta$D, we calibrated the analysers down to 50

ppmv (as stated in the text l. 142), so that the range of calibration validity for both isotopes is the same.

33: Line 314: Could you please confirm the Eq. 2? The correction seems linearly depend on measured humidity.

We confirm the correctness of Equation 2. As mentioned above (comments n°19 and 21), we do not use the calibrated humidity to correct for the humidity-isotope response, but the measured humidity instead. This is to prevent incorporating errors from the humidity calibration. The humidity-isotope response correction is therefore independent of the humidity calibration.

34: Line 320: Could you simplify the legend and the figure? The hatch is nearly invisible when printed, do you really need to show the ±2 sigma? You can use one color for each instrument and one shape for each lab standard. Currently it is very confusing.

We need to show the ±2 sigma to show the outliers. However, we agree that the figure is confusing, we have updated it to the following (including modifications from Reviewer 1 comment n°12):

[Figure]

Figure 4: Results of the regular [...]. Panels (b) and (c) show the measured isotopic ratios, corrected for

the humidity-isotope response (Sect. 2.2.2 and 3.1.2), by both analysers during each calibration as a deviation of the mean over the whole period ($\Delta\delta^* = \delta^*_{humcorr} - \overline{\delta^*_{humcorr}}$, * is for any isotope). [...]

35, Line 330: What if you plot the humidity generated by LHLG against the deviations in d18O? Do you have records of room temperature?

Plotting the humidity generated by the LHLG versus the deviation in $\delta^{18}O$ and $\delta D$ does not show a clear correlation / dependence between the two (see additional figure below). In addition, the LHLG heats during the calibration, which should limit the effect of the room temperature. However, variations of the latter might still affect the stability of the calibration and therefore the results. Unfortunately, this is difficult to check since we do not have any records of the room temperature.

[Figure]

36, Line 342: Do you mean a small spectral window? Could you provide the frequency range for $\delta^{18}O$ and $\delta D$? Could it also result in the different performance we saw in Fig 3? While for many analyses, people seem to pick either $\delta D$ or $\delta^{18}O$ arbitrarily, here it seems that for observations, the relative uncertainty is smaller for $\delta D$ and the application range is larger for $\delta D$ than $\delta^{18}O$.

We modified the text to explain what we meant here (l. 340-342, in bold):

[...] with the results from Lauwers et al. (2025). We observe that for both laboratory standards, the variations in $\delta D$ over the period are higher for the AP2E analyser than Picarro. One reason that could explain this difference is that the OF-CEAS technique used in AP2E spectrometers is particularly sensitive to noise associated with optical absorption, compared to the CRDS technique used in Picarro spectrometers (Lauwers et al., 2025). This effect is more visible when the absorption peak is very close to the baseline: for example at low humidity, or when looking at the deuterium absorption peak which shows an amplitude one order of magnitude smaller than the $^{18}O$ peak. We further exclude [...]

The spectral window includes HDO, $H_2^{18}O$ and $H_2^{16}O$ peaks, with a frequency range of approx. 0.1 nm around a central frequency of 1388.85 nm (for more details, see Figure 1 in Lauwers et al., 2025). Concerning Figure 3, the "high humidity" offset for the $\delta^{18}O$ calibration curve observed on the AP2E instrument (from 500 to 1100 ppmv) is well known and explained by the absorption spectrum baseline which is influenced by absorption peaks situated outside the spectral window, and affect more markedly $H_2^{18}O$ than HDO due to their different position in the spectrum. This correction does not affect the application range for $\delta D$ and $\delta^{18}O$ in the high humidity region. So in general for most observations that are situated above 500 ppmv, there is no difference in picking $\delta D$ or $\delta^{18}O$. But, we agree that in the very low humidity region encountered here, calibration uncertainty may add up to the global uncertainty. Figure 3 shows indeed a larger low humidity divergence for $8*\delta^{18}O$ than $\delta D$, indicating that low humidity correction may be less biased for $\delta D$ (in particular if extrapolation is used).

37, Line 361: Could you provide RMSE here? How can you confirm that the vapour flux generated by LHLG has an isotope ratio as per the lab standard? In the legend, y=a±bx + c±d, not y=ax±b + c±d.

We have added the RMSE on Figure 5:

[Figure]

Figure 5: Humidity-isotope corrected ratios vs true isotopic ratios ($\delta^{18}O$ in panel (a) and $\delta D$ in panel (b)) of two laboratory standards (FP5 and NEEM) for both Picarro and AP2E analysers. In both panels, the smaller coloured markers represent all selected calibrations and the larger coloured markers the average isotopic ratio of all selected calibrations (whiskers represent one standard deviation). The coloured lines show the linear regressions between the true and humidity-corrected isotopic ratios using two laboratory standards. On each panel the root mean square error (RMSE, in ppmv) calculated between the two standards is shown.

The new figure legend has now the correct linear function description (y=(a±b)x + (c±d)). The LHLG instrument was designed specifically to generate humid air with the same isotopic composition of the liquid standard injected in the system (see details in Kerstel, 2021 and Leroy-Dos Santos et al., 2021).

38, Line 382: Is it because AP2E is over-tuned for low $\delta$D values?

No, the AP2E analyser is not tuned at all, everything is done in post-processing. In the Picarro, some corrections are already included in the instrument (output raw data already include some calibrations / corrections which we do not have control on). This can explain why both instruments need a different absolute calibration.

39, Line 410: Could you quantify the completeness of the dataset over a specific period?

We have added the percentage of 1-hourly missing data in the text, as follows (l. 410):

Note that the time series are not continuous, with interruptions due to calibration periods, maintenance work on the instruments or electrical shutdowns. Missing data represents 21% of the overall dataset (December 6th 2023 to March 14th 2024).

40, Line 413: Could you provide the root mean squared difference, mean difference, R squared values, and mean absolute difference between raw and calibrated humidity, $\delta$D, $\delta^{18}$O for each analyser, as well as the statistics between two analysers? The differences could be quite large if you include periods of low humidity. Could you determine a threshold to do the calculation and also as a recommendation on which range is more reliable (e.g. for model-data comparison)?

[Figure]

Figure 6: Time series (December 6th 2023 to March 14th 2024) of the atmospheric humidity (in ppmv, panel (a)), $\delta^{18}O$ (in ‰, panel (b)), $\delta D$ (in ‰, panel (c)) and d-excess (in ‰, panel (d)) measured by the Picarro (green lines) and AP2E (blue lines) analysers. In panels (b), (c) and (d), the green and blue shaded areas correspond respectively to $\sigma(h)$ (Sect. 2.2.4) of the Picarro and AP2E analysers. In all four panels, the dashed lines correspond to the raw data given by the spectrometers and the plain lines correspond to the corrected and calibrated data (see Sect. 2.2 and 3.1). The grey hatched area marks the period from January 11th to January 15th shown in Fig. 7. In all panels, the root mean square difference (RMSD, in ppmv), mean difference (MD, calibrated – raw or Picarro – AP2E, in ppmv), mean absolute difference (MAD, in ppmv) and the squared Pearson correlation coefficient ($R^2$) between the raw and calibrated datasets (green for Picarro, blue for AP2E) and between the calibrated time series of the two analysers (grey) are shown.

To include this information in the text, we have modified the text as follows (l. 428-470, in bold):

**The raw humidity measured by both analysers show the same variations over the whole period (Fig. 6a and 7a), except for a bias already identified in Sect. 3.1.1. After the calibration against the independent humidity sensor, the humidities are in excellent agreement over the whole period (RMSD = 4.5 ppmv, MD = -1.0 ppmv, MAD = 2.7, $R^2$ = 1.0, Figure 6a).** The calibrated humidities are showing the same diurnal variations for both

analysers, synchronous with the temperature diurnal cycle on site (Fig. 7a). In addition, both instruments record the decrease of the humidity from the beginning of February, coinciding with the onset of the winter at Dome C (Fig. 6a).

Contrary to the humidity, the calibration of the raw data has a significant effect on the $\delta^{18}O$ time series of both analysers. For the AP2E analyser, the calibration of the raw $\delta^{18}O$ time series shifts it towards higher values (Fig. 6b and 7b), **with a mean difference of 9.2‰ over the whole period between the raw and calibrated time series (Fig. 6b)**. This shift is expected from the absolute calibration curve (Sect. 3.1.3). The amplitude of the diurnal cycle is also slightly reduced after applying the calibration (Fig. 7b), due to the humidity- $\delta^{18}O$ response of the analyser (i.e. positive correction for low humidities and negative correction for high humidities, Sect. 3.1.2). **For the Picarro analyser, the raw and calibrated $\delta^{18}O$ time series show a mean difference of -9.3‰, which is mostly due to the part of the time series from beginning of February onwards (Fig. 6b).** The amplitude of the diurnal cycle is larger after calibration, as expected from the humidity- $\delta^{18}O$ response of the Picarro which shows negative correction for lower humidities (Fig. 3a, Sect. 3.1.2). This is further visible on the period from the end of January onwards, where the diurnal cycles show an opposite behaviour between the raw and calibrated data: the raw data is in opposite phase to the humidity (minimum $\delta^{18}O$ associated with maximum humidity) and the calibrated data is in phase with the humidity (minimum $\delta^{18}O$ associated with minimum humidity). This is an effect of the large humidity- $\delta^{18}O$ response of the Picarro at low humidities (Fig. 3a, Sect. 3.1.2).

Compared to $\delta^{18}O$, the raw and calibrated $\delta D$ time series from both instruments are rather similar, at least during the period where the humidity is above 200 ppmv (mid-December to end of January, Fig. 6c). **The calibration of both analysers modifies the average $\delta D$ values (mean difference of 7.4‰ for the Picarro and 19.1‰ for the AP2E analyser over the whole period, Fig. 6c).** The calibration of the $\delta D$ time series does not affect the amplitude of the diurnal cycle for neither analyser (Fig. 7c). Both raw $\delta D$ time series compare relatively well from mid-December to the end of January (dashed lines in Fig. 6c), with the same in-phase relationship between $\delta D$ and the mixing ratio as for the calibrated $\delta^{18}O$ time series. This in-phase relationship between $\delta D$ and the humidity is preserved after calibration (plain lines in Fig. 6c and 7c).

**There is a good agreement between the $\delta^{18}O$ calibrated time series from the AP2E and Picarro analysers over the period, until they start to diverge mid-February (Fig. 6b). Considering the entire period, the two $\delta^{18}O$ calibrated time series show a mean absolute difference of 3.6‰, which is within the range of uncertainties of the calibrated time series (shaded areas in Fig. 6b), and a squared Pearson correlation coefficient $R^2$ of 0.58 (Fig. 6b). When considering only the period before mid-February, the mean absolute difference is reduced to 1.8‰ and $R^2$ is improved to 0.8 (not shown). The good agreement between the two analysers confirms that the calibration is valid for the range of humidities encountered over this period.**

As for $\delta^{18}O$, we observe that the calibrated $\delta D$ time series from both instruments agree well between mid-December to mid-February, **when similarly to $\delta^{18}O$ they start to diverge (Fig. 6c). Over the entire period, the mean absolute difference between the two calibrated $\delta D$**

**time series is 22.1‰, which is also within the uncertainty of both calibrated time series (shaded areas in Fig. 6c), and $R^2$ is 0.59 (Fig. 6c). When considering only the period before mid-February, the mean absolute difference is reduced to 10.7‰ and $R^2$ is improved to 0.85 (not shown).**

Finally, the raw time series of d-excess are very different between the two analysers (Fig. 6d and 7d). However, after the calibration of both analysers, the two d-excess time series are comparable within their uncertainty range (Fig. 6d and 7d). As for $\delta^{18}O$ and $\delta D$, the calibrated d-excess time series of the two analysers diverge from mid-February onwards (Fig. 6d).

The divergence **from mid-February** in both $\delta^{18}O$ and $\delta D$ between the two instruments is probably due to the increase of instantaneous measurement noise of the analysers when the humidity decreases. It is also related to the difficulty of calibrating the instruments for very low humidity levels (Sect. 3.1.2). This is reflected in the uncertainty of the measurements which increases significantly for both instruments from mid-February onwards (Fig. 6b and c), when the mixing ratio is consistently below 200 ppmv (Fig. 6a). **In addition, as stated above, the mean absolute difference and $R^2$ values calculated between the two calibrated time series are improved when considering the period before mid-February. We therefore restrict the comparison between the observations and the model in Sect. 3.3 to the period before mid-February, and make available in public access only this part of the dataset in Landais et al. (2024b) for future work.**

41, Line 487: Could you please provide statistics as mentioned before? Is it possible to compare temperature and provide it in the dataset as well?

We have modified Figure 8 to include the statistics mentioned in the reviewer's previous comment, as follows:

[Figure]

Figure 8: Time series (December 6th 2023 to February 14th 2024) of the atmospheric humidity (in ppmv, panel (a)), $\delta^{18}O$ (in ‰, panel (b)), $\delta D$ (in ‰, panel (c)), and d-excess (in ‰, panel (d)) measured (and calibrated) by the Picarro analyser (green lines), the AP2E (blue lines) analysers, and modelled by LMDZ6-iso (grey lines). In panels (b), (c) and (d), the green and blue shaded areas correspond respectively to σ(h) (Sect. 2.2.4) of the Picarro and AP2E analysers. In all four panels, the grey hatched area marks the period from January 1st to 11th 2024 shown in Fig. 9 (same period as in Fig. 7). In panel (a), the light brown area marks the period from December 16th to 20th 2023 (period when the modelled and observed humidities differ). In all panels, the root mean square difference (RMSD, in ppmv), mean difference (MD, model – observations, in ppmv), mean absolute difference (MAD, in ppmv) and the squared Pearson correlation coefficient ($R^2$) between the measured and modelled time series are shown (green for Picarro vs LMDZ, blue for AP2E vs LMDZ).

For clarity on Figure 8, we decided to not add the time series of temperature. However, we did calculate the aforementioned statistics between the temperature measured by the AWS and modelled by LMDZ over the period from December 6th to February 14th: RMSD = 3.5°C, MD = -0.6°C, MAD = 2.8°C, and $R^2 = 0.81$.

To include this information in the manuscript, we have modified the text as follows (l. 490-523, in bold):

The comparison of the humidity modelled by LMDZ6-iso and measured by both analysers show an overall good agreement **albeit a positive bias in the model (MD = 36.0 ppmv and $R^2$ = 0.82 compared to Picarro; MD = 29.0 ppmv and $R^2$ = 0.83 compared to AP2E, Figure 8a)**, including for the amplitude of the observed diurnal cycle (Fig. 8a and 9a). However, during some specific periods, the model shows higher humidity levels than what is observed, especially during the nighttime (e.g. December 16th to 20th, light brown area in Fig. 8a). **Contrary to the humidity, the air temperature modelled by LMDZ does not exhibit any mean bias compared to the air temperature measured by the local AWS (MD = -0.6°C and MAD = 2.8°C, not shown).**

Although the model reproduces the observed in-phase relationship between $\delta^{18}O$ and the humidity, the comparison between the modelled and observed $\delta^{18}O$ shows a poorer agreement than for humidity. Firstly, the modelled $\delta^{18}O$ shows an **overall positive bias over the whole period, with a mean difference of 4.9‰ compared to the Picarro analyser and 3.0‰ compared to the AP2E analyser (Fig. 8b).** Secondly, the amplitude of the diurnal cycle modelled by LMDZ6-iso is overall larger than in the observations (Fig. 8b). Over the period January 11th to January 15th 2024 (Fig. 9b), the amplitude of the mean diurnal cycle in $\delta^{18}O$ modelled by LMDZ6-iso is 10.9‰ (from -70.9 to -60.0‰, not shown), higher than the one from both the Picarro analyser (5.7‰, Sect. 3.2) and the AP2E analyser (4.7‰, Sect. 3.2).

The same patterns are observed for $\delta D$. The modelled $\delta D$ also shows an overall mean positive bias compared to the observations, **with a mean difference of 27.3‰ compared to the Picarro analyser and 19.4‰ compared to the AP2E analyser (Fig. 8c).** The amplitude of the diurnal cycle is also larger in LMDZ6-iso than in the observations (Fig. 8c). Between January 11th and January 15th 2024 (Fig. 9c), the mean diurnal amplitude modelled by LMDZ6-iso is 69.0‰ (from -515.8 to -446.8‰, not shown), which is higher than the observed one (34.9‰ for Picarro analyser, 29.5‰ for AP2E analyser, Sect. 3.2).

Lastly, due to the biases identified for $\delta^{18}O$ and $\delta D$, the d-excess modelled by LMDZ6-iso also shows some discrepancies with the observations. The model shows an overall negative bias compared to the observations, **with a mean difference over the whole period of 12.1‰ compared to the Picarro analyser and of 4.6‰ compared to the AP2E analyser (Fig. 8d).** The comparison of the amplitudes of the diurnal cycle is less conclusive than for $\delta^{18}O$ and $\delta D$, due to the large uncertainties associated with the observations (Fig. 9d). However, we observe that the model still correctly captures the observed anti-phase relationship between d-excess and $\delta^{18}O$ (or $\delta D$), with a maximum d-excess when $\delta^{18}O$ is minimal, i.e. during the night, and a minimum d-excess when $\delta^{18}O$ is maximal, i.e. during the day (Fig. 9d).

42, Line 522: The anti-phase relationship may be an artefact due to the linear definition of dxs (see previous reference Dütsch et al. 2017).

We agree with the reviewer's comment. Here in the text we are only pointing out that the model reproduces the anti-phase relationship between $\delta^{18}O$ ($\delta D$) and d-excess, regardless if this relationship is an artefact of the linear definition of d-excess, as pointed out by the reviewer, or the "true" behaviour of d-excess compared to $\delta^{18}O$ and $\delta D$.

43, Line 547: what are the isotope compositions of surface snow at the station in the model (and in reality)? How does it compare to vapour isotopes and if sublimation occurs with equilibrium fractionation, is the sublimation flux more depleted or enriched than the vapour fluxes?

We already provide the difference between the isotopic composition of the surface snow at the station and in the model later in the text (l. 564), however we modified the text to explicitly provide the values, as follows (l. 562-566):

This overall bias in the modelled vapour isotopic composition could be explained by the isotopic composition of the snow in LMDZ6-iso, which might differ significantly from the actual snow surface at Dome C. Indeed, the average isotopic composition of the surface snow in LMDZ6-iso over the period December 2023 – January 2024 (-48.5‰ in $\delta^{18}O$, -369‰ in $\delta D$) is higher (+1‰ in $\delta^{18}O$ and +19‰ in $\delta D$) than the summertime average isotopic composition of the surface snow at Concordia Station (-49.3‰ in $\delta^{18}O$, -388‰ in $\delta D$, average over all December and January months between 2017 and 2021, Ollivier et al., 2025).

We calculated the isotopic composition of the sublimation flux in equilibrium conditions, using the summertime daily cycle maximum conditions in the model LMDZ and the observations (equilibrium fractionation coefficients from Merlivat and Nief, 1967 and Majoube, 1970, and surface temperature of -31°C):

- For LMDZ: $\delta^{18}O_{snow}$* = -48.5‰ $\rightarrow$ $\delta^{18}O_{flux}$ = -68.0‰ ; and $\delta^{18}O_{vapour}$** $\approx$ -60‰
- For the observations: $\delta^{18}O_{snow}$* = -49.3‰ $\rightarrow$ $\delta^{18}O_{flux}$ = -68.7‰ ; and $\delta^{18}O_{vapour}$** $\approx$ -66‰

*mean summertime (December-January) $\delta^{18}O$ in the surface snow (see modified text above)

**maximum $\delta^{18}O$ during typical diurnal cycle at Concordia station (values taken from Figure 9b)

In the case of the LMDZ model, including equilibrium isotopic fractionation during sublimation would contribute to lowering the isotopic composition of the vapour itself ($\delta^{18}O_{flux} \ll \delta^{18}O_{vapour}$, by $\approx$ 8‰).

Dutrievoz, N., Agosta, C., Risi, C., Vignon, É., Nguyen, S., Landais, A., Fourré, E., Leroy-Dos Santos, C., Casado, M., Masson-Delmotte, V., Jouzel, J., Dubos, T., Ollivier, I., Stenni, B., Dreossi, G., Masiol, M., Minster, B., and Prié, F.: Antarctic Water Stable Isotopes in the Global Atmospheric Model LMDZ6: From Climatology to Boundary Layer Processes, JGR Atmospheres, 130, e2024JD042073, https://doi.org/10.1029/2024JD042073, 2025.

Genthon, C., Piard, L., Vignon, E., Madeleine, J.-B., Casado, M., and Gallée, H.: Atmospheric moisture supersaturation in the near-surface atmosphere at Dome C, Antarctic Plateau, Atmos. Chem. Phys., 17, 691–704, https://doi.org/10.5194/acp-17-691-2017, 2017.

Kerstel, E.: Modeling the dynamic behavior of a droplet evaporation device for the delivery of isotopically calibrated low-humidity water vapor, Atmos. Meas. Tech., 14, 4657–4667, https://doi.org/10.5194/amt-14-4657-2021, 2021.

Ollivier, I., Steen-Larsen, H. C., Stenni, B., Arnaud, L., Casado, M., Cauquoin, A., Dreossi, G., Genthon, C., Minster, B., Picard, G., Werner, M., and Landais, A.: Surface processes and drivers of the snow water stable isotopic composition at Dome C, East Antarctica – a multi-dataset and modelling analysis, The Cryosphere, 19, 173–200, https://doi.org/10.5194/tc-19-173-2025, 2025.

Steen-Larsen, H. C., Johnsen, S. J., Masson-Delmotte, V., Stenni, B., Risi, C., Sodemann, H., Balslev-Clausen, D., Blunier, T., Dahl-Jensen, D., Ellehøj, M. D., Falourd, S., Grindsted, A., Gkinis, V., Jouzel, J., Popp, T., Sheldon, S., Simonsen, S. B., Sjolte, J., Steffensen, J. P., Sperlich, P., Sveinbjörnsdóttir, A. E., Vinther, B. M., and White, J. W. C.: Continuous monitoring of summer surface water vapor isotopic composition above the Greenland Ice Sheet, Atmos. Chem. Phys., 13, 4815–4828, https://doi.org/10.5194/acp-13-4815-2013, 2013.

Vignon, É., Raillard, L., Genthon, C., Del Guasta, M., Heymsfield, A. J., Madeleine, J.-B., and Berne, A.: Ice fog observed at cirrus temperatures at Dome C, Antarctic Plateau, Atmos. Chem. Phys., 22, 12857–12872, https://doi.org/10.5194/acp-22-12857-2022, 2022.

Weng, Y., Touzeau, A., and Sodemann, H.: Correcting the impact of the isotope composition on the mixing ratio dependency of water vapour isotope measurements with cavity ring-down spectrometers, Atmos. Meas. Tech., 13, 3167–3190, https://doi.org/10.5194/amt-13-3167-2020, 2020.

---

## Author Response (AR3)

**Response to topic editor**

We thank the editor for their comments on the revised version of the manuscript. We have addressed all comments below and implemented the changes in a new revised version of the manuscript. The line numbers indicated below correspond to the latest version of the manuscript (revised version after implementation of reviewer's comments). Note that we have also updated the reference to the dataset (now published on PANGAEA).

Black: editor's comment (*italic*: authors response to initial reviewer's comments)

Blue: author's response Green: revised text

**Reviewer 1:**

**You replied to a comment:**

> We agree with the reviewer's comment that we used two standards mostly outside of the range of measured isotopic ratios. However, the standard mentioned here by the reviewer (VSAEL) was not yet made available when the calibrations of the instruments were performed in the field. We therefore used the most depleted standard available to us at that time, which was the homemade FP5 standard.

There will be other readers wondering about this. Can you make a note on this in the text with the release date of the VSAEL?

To be exact, the standard VSAEL was existing but not available for calibrations in the field when our instruments were deployed (reserved for specific use in the laboratory). We included a note in the text as follows (l. 163, in bold):

[...] -257.2‰). Note that the very depleted standard VSAEL was not available for calibrations in the field when our instruments were deployed. The calibrations [...]

**You replied to a comment:**

> We attribute the different behaviour of the two analysers to the humidity levels. At these low humidities (below 50 ppmv, which is most of the time in March when the two instruments show very different behaviour), the humidity-isotope response calibrations are not constrained for either analyser (lowest calibration point is 50 ppmv, as explained in Sect. 2.2.2). Therefore neither instrument is really reliable at these humidities because the calibration is extrapolated. This is also the reason why we discard this period for the observation-model comparison. However, it seems that this particular Picarro analyser would be best at capturing the atmospheric water vapour isotopic composition at these low humidities, as shown in Figure 6b and c where the diurnal cycle in  $\delta^{18}O$  and  $\delta D$  measured by the Picarro follows the diurnal cycle in the atmospheric humidity (minimum  $\delta^{18}O$  and  $\delta D$  associated with minimum humidity), which is what we expect. In contrast, the AP2E shows an opposite phase between  $\delta^{18}O$  and  $\delta D$  during March. Nevertheless, this does not permit concluding on the ability of the AP2E or Picarro analyser to measure at very low humidity levels, since the data

shown here are only calibrated down to 50 ppmv. Further efforts should be made to better constrain the calibration in order to use the instruments at very low humidities.

A condensed version of this analysis should be added to the text. This could be presented as a qualitative extension of your comparison for when both instruments are outside their calibration range and therefore when quantitative comparison is not possible.

We added a condensed version of this analysis to the manuscript (after 1. 486, in bold below):

[...] and make available in public access only this part of the dataset in Landais et al. (2024b) for future work.

After mid-February, it seems that this particular Picarro analyser best captures the atmospheric water vapour isotopic composition at these low humidities. As shown in Figure 6b and c, the diurnal cycle in  $\delta^{18}O$  and  $\delta D$  measured by the Picarro follows the diurnal cycle in the atmospheric humidity (minimum  $\delta^{18}O$  and  $\delta D$  associated with minimum humidity), which is what we expect. In contrast, the AP2E analyser shows an opposite phase between the humidity and the  $\delta$ -values. Nevertheless, this does not permit concluding on the ability of the AP2E or Picarro analyser to measure at very low humidity levels, since the data shown here are only calibrated down to 50 ppmv (Sect. 2.2.2). Further efforts should be made to better constrain the calibration in order to use the instruments at very low humidities.

**Reviewer 2:**

**Comment 9, you replied:**

> Yes, this is a correct observation. We modified the text to make our point clearer as follows (l. 30–36, in bold):

In polar ice cores,  $\delta^{18}O$  and  $\delta D$  have traditionally been interpreted as a temperature proxy based on empirical relationships between the mean annual temperature and the isotopic composition of snow samples (e.g. Johnsen et al., 1992; Jouzel et al., 2007; Lorius et al., 1979). Alongside, d-excess has been interpreted as a proxy for climatic conditions at the evaporative source region (e.g. Landais et al., 2021; Stenni et al., 2010; Uemura et al., 2008; Vimeux et al., 1999). However, an increasing number of studies have shown that the isotopic composition ( $\delta^{18}O$ ,  $\delta D$ , and d-excess) of the snow surface and deeper layers in the snowpack is affected by post-depositional processes at the ice sheet surface (e.g. Casado et al., 2018, 2021; Ollivier et al., 2025; Steen-Larsen et al., 2014; Town et al., 2024; Zuhr et al., 2023).

I agree with the reviewer that it is not clear why there is opposition between the two parts of your statement (marked by "however"). If the relation between isotopic composition measured in ice cores and temperature was derived empirically, then it already includes the postdepositional processes and there is no reason to present the postdepositional processes as contradicting established proxies. If, on the contrary, the studies you mention highlight shortcomings in established relationships stemming from postdepositional processes (dating uncertainty, removal or migration of peaks in isotopic profiles...), then please be more specific about what they are. It is a good opportunity to introduce how

your high-quality surface measurements are necessary to better understand the link between atmospheric state and ice core records.

We agree with both the reviewer and the editor that the opposition term "however" is misleading here. We have replaced it by the term "In the last decade" (1. 33-35, in bold below):

[...] evaporative source region (e.g. Landais et al., 2021; Stenni et al., 2010; Uemura et al., 2008; Vimeux et al., 1999). In the last decade, an increasing number of studies have shown that the isotopic composition ( $\delta^{18}$ O,  $\delta$ D, and d-excess) [...]

The two following sentences then emphasize on the importance of measurements of the atmospheric water vapor isotopic composition.

**Comment 10, you replied:**

> The method used in Leroy-Dos Santos to estimate the uncertainty is different from the method we used in our manuscript, so it is a bit delicate to compare directly. Nevertheless, they estimate the uncertainty on  $\delta^{18}O$  to be approximately 2.5% when the humidity is maximal (between 400 and 600 ppmv, their Figure 7) and close to 4.5% when the humidity is minimal (around 200 ppmv, their Figure 7). In our study, we find the uncertainty on the  $\delta^{18}O$  measured by the Picarro to be between 0.3% at 1000 ppmv (approximate maximum value during the studied time period) and 1.5% at 200 ppmv. Both values were calculated with Equation 4 and the values of  $\sigma(1)$ + in Table 4.

Has this been added to the main text?

We added this information to the text (1. 398, in bold below):

[...] both analysers presented along the data in the following section. For comparison, the uncertainty on  $\delta^{18}O$  estimated by Leroy-Dos Santos et al. (2021) is approximately 2.5‰ when the humidity is maximal (between 400 and 600 ppmv, their Fig. 7) and close to 4.5‰ when the humidity is minimal (around 200 ppmv, their Fig. 7). Although our estimation method is different, we find here the uncertainty on the  $\delta^{18}O$  measured by the Picarro to be between 0.3‰ at 1000 ppmv (approximate maximum value during the studied time period) and 1.5‰ at 200 ppmv. Both values were calculated with Equation 4 and the values of  $\sigma_{i,drift}$  in Table 4.

**Comment 16. Websites' URL:**

Footnotes should be avoided, please make it a reference to a webpage: <a href="https://www.earth-system-science-data.net/submission.html#references">https://www.earth-system-science-data.net/submission.html#references</a>

We modified the text to the following (1.91-100, in bold below):

A Picarro L2130-i analyser (Picarro Inc., CRDS measurement technique, Picarro analyser hereinafter; **Picarro, 2025**) [...] In parallel to the Picarro analyser, a prototype (non commercially available) of a

AP2E ProCeas analyser (AP2E Inc., OF-CEAS measurement technique, AP2E analyser hereinafter; **AP2E**, **2025**) [...]

And added the references to the reference list as follows:

Picarro:

https://www.picarro.com/environmental/products/l2130i\_isotope\_and\_gas\_concentration\_analyzer, last access: 19 September 2025

AP2E: https://www.ap2e.com/en/our-gas-analyzers/proceas/, last access: 19 September 2025

**Comment 21, you replied:**

> We conducted 9 individual calibrations, each associated with a single humidity level (the humidity does not vary within each calibration). Each calibration lasts about two hours, and we average the last 10 minutes to get each data point. This way, we minimize the memory effect by ensuring that potential residual water from previous calibrations has been evacuated from the system and that the humidity level has reached a stable state. In addition, compared to measurement uncertainty, the memory effect is negligible at 200 ppmv. Nevertheless, memory effect is the reason why we do not go below 50 ppmv for the calibration steps, as the residual water is not negligible. In Figure 3, the calibration humidities are different for both instruments because we show the raw humidity. As explained in our answer to comment n°19 above, we do not calibrate the isotopes with the calibrated humidity but with the measured humidity, to prevent incorporating an additional error. Therefore both instruments, although the target calibration humidity is the same, will not measure the same humidity and not the precise target humidity (that is why it also appears as random values).

We have added the information about the number of calibrations in the text as follows (l. 135): [...] performing one series of nine calibrations in January 2024.

Please add information about the calibration duration and the assumption that memory effect is negligible at 200 ppmv to the text.

**We added this information to the text (l. 146, in bold below):**

[...] The calibration steps were performed from high to low humidity (humidities ranging from 1100 to 50 ppmv). Each calibration lasts approximately two hours, and the data point correspond to the average of the last 10 minutes, in order to minimize the memory effect. We assume that at 200 ppmv, the memory effect is negligible compared to the measurement uncertainty. The humidity levels [...]

**Comments 31 & 32:**

Is there a clear statement somewhere in the text saying that the humidity-isotope response is

known/expected to be instrument-dependent?

**Yes, this statement is made 1. 134-135:**

"This humidity-isotope response is instrument-specific (e.g. Steen-Larsen et al., 2013) and is dependent on the isotopic composition of the laboratory standard used to perform the calibrations (e.g. Lauwers et al., 2025; Weng et al., 2020)."

**And 1. 282-284:**

"The difference in humidity-isotope response of the two Picarro analysers (HIDS2319 and HIDS2308) is not surprising since different spectrometers will have a different humidity-isotope response (e.g. Steen-Larsen et al., 2013)."

**Comment 34:**

I agree that the  $2\sigma$  ranges are hard to read. Please replace them with vertical whiskers on the right-hand side of each panel. Please add a red marker for outliers in the legend. The legend of the three panels could be merged and placed at the top of the figure.

We have modified Figure 4 and its caption to include the editor's comment:

Figure 4: Results of the regular calibrations performed with two laboratory standards (FP5 and NEEM) between

January 11th and June 6th with the new version of the LHLG (description in Sect. 2.2.2 and 2.2.3). Panel (a) shows the humidity measured by both analysers during each calibration. The red markers show the calibrations that were discarded (outside of two standards deviations around the mean humidity, **indicated by the vertical bars on the right-hand side**). Panels (b) and (c) show the measured isotopic ratios, corrected for the humidity-isotope response (Sect. 2.2.2 and 3.1.2), by both analysers during each calibration as a deviation of the mean over the whole period ( $\Delta \delta^* = \delta_{i,humcorr} - \overline{\delta_{i,humcorr}}$ , subscript  $_i$  is for each isotope species). The isotopic ratios of each calibration are corrected for the isotope-humidity response of each analyser. In panels (b) and (c), only the accepted calibration from panel (a) are shown. The red markers show the calibrations that are discarded in a second step (outside of two standard deviations around the mean isotopic ratio, **indicated by vertical bars on the right-hand side**).

**Comment 37:**

The RMSE annotations seem quite out of place. Either mention them in the text only or alternatively: - move the RMSE to the legend where instrument name, linear equation, and RMSE are each on a new line

- potentially move the legend outside (e.g. on top of) the plotting area and reduce the size of the panel Please add panel labels.

We have modified Figure 5 and its caption to include the editor's comment:

Figure 5: Humidity-isotope corrected ratios vs true isotopic ratios ( $\delta^{18}$ O in panel (a) and  $\delta$ D in panel (b)) of two laboratory standards (FP5 and NEEM) for both Picarro and AP2E analysers. In both panels, the smaller coloured markers represent all selected calibrations and the larger coloured markers the average isotopic ratio of all selected calibrations (whiskers represent one standard deviation). The coloured lines show the linear regressions between the true and humidity-corrected isotopic ratios using the two laboratory standards.

**Comment 40:**

I believe there is now too much information in Figure 6 and statistics look like small tables inserted in a plot. Please make a clearer distinction between quantifying the impact of calibration (which should be done before section 3.2) and presenting your final calibrated values with their derived uncertainties

(which is section 3.2 and Figure 6). Prior to section 3.2, you should present the raw vs calibrated statistics either in a table or in scatter plots (e.g. additional panels in Figure 5) and discuss them there. When coming to section 3.2 and Figure 6, the focus should be on the temporal evolution of the calibrated values, their uncertainties, and how they differ for the two instruments. Again, the grey statistics in Figure 6 should either be in a table or shown on a scatter plot (potentially as additional, squared panels displayed on the right-hand side of each time series plot in Figure 6), so there is no doubt about which series are being compared.

We argue that the presentation and comparison of the raw and calibrated data should still be included in Section 3.2, since Section 3.1 is presenting the results of the calibration steps themselves. However, we do agree that is should be clearer when we discuss the effect of the calibration on the data and the instrument inter-comparison. In addition, we agree that the statistics inserted in Figure 6 makes the figure hard to read. To accommodate for the different issues, we modified the manuscript by:

- Adding two sub-sections in Section 3.2:
  - o 3.2.1 Calibration effect on measured time series includes text from 1. 438 to 463
  - o 3.2.2 Instrument inter-comparison includes text from 1. 465 to 495
- Removing the statistics inserted in Figure 6 and adding two tables to the text:
  - Table 5 in new Section 3.2.1 with statistics for the comparison raw calibrated time series
  - Table 6 in new Section 3.2.2 with statistics for the instrumental inter-comparison of the two calibrated time series

**The Section 3.2 is now the following (excluding Figures 6 and 7, bold indicates new/modified text):**

**3.2 Time series of the water vapor isotopic composition**

Figure 6 shows the evolution of the atmospheric humidity,  $\delta^{18}$ O,  $\delta$ D and d-excess measured by both laser spectrometers between December 2023 and March 15th 2024. Figure 7 shows a focus on a four-day period in January 2024 (corresponding to the grey hatched area in Fig. 6). Note that the time series are not continuous, with interruptions due to calibration periods, maintenance work on the instruments or electrical shutdowns. Missing data represents 21% of the overall dataset (December 6th 2023 to March 15th 2024). The air temperature shown in Fig. 7 is measured at 1.5 m above the surface by an Automatic Weather Station (AWS) installed in the vicinity of Concordia station (Grigioni et al., 2022). The comparison of the raw and calibrated time series from the Picarro and AP2E analysers is described in Sect. 3.2.1 and statistics over the whole period are summarized in Table 5. The instrument inter-comparison of the calibrated time series from the two analysers is described in Sect. 3.2.2 and statistics over the whole period are summarized in Table 6.

**3.2.1 Calibration effect on measured time series**

The raw humidity measured by both analysers show the same variations over the whole period (Fig. 6a and 7a), except for a bias already identified in Sect. 3.1.1. After the calibration against the independent humidity sensor, the humidities are in excellent agreement over the whole period (**Table 5**). The calibrated humidities are showing the same diurnal variations for both analysers, synchronous

with the temperature diurnal cycle on site (Fig. 7a). In addition, both instruments record the decrease of the humidity from the beginning of February, coinciding with the onset of the winter at Dome C (Fig. 6a).

Contrary to the humidity, the calibration of the raw data has a significant effect on the  $\delta^{18}$ O time series of both analysers. For the AP2E analyser, the calibration of the raw  $\delta^{18}$ O time series shifts it towards higher values (Fig. 6b and 7b), with a mean difference of 9.2% over the whole period between the raw and calibrated time series (Table 5). This shift is expected from the absolute calibration curve (Sect. 3.1.3). The amplitude of the diurnal cycle is also slightly reduced after applying the calibration (Fig. 7b), due to the humidity- $\delta^{18}$ O response of the analyser (i.e. positive correction for low humidities and negative correction for high humidities, Sect. 3.1.2). For the Picarro analyser, the raw and calibrated  $\delta^{18}$ O time series show a mean difference of -9.3% (**Table 5**), which is mostly due to the part of the time series from beginning of February onwards. The amplitude of the diurnal cycle is larger after calibration, as expected from the humidity- $\delta^{18}$ O response of the Picarro which shows negative correction for lower humidities (Fig. 3a, Sect. 3.1.2). This is further visible on the period from the end of January onwards, where the diurnal cycles show an opposite behaviour between the raw and calibrated data: the raw data is in opposite phase to the humidity (minimum  $\delta^{18}$ O associated with maximum humidity) and the calibrated data is in phase with the humidity (minimum  $\delta^{18}$ O associated with minimum humidity). This is an effect of the large humidity- $\delta^{18}$ O response of the Picarro at low humidities (Fig. 3a, Sect. 3.1.2).

Compared to  $\delta^{18}$ O, the raw and calibrated  $\delta D$  time series from both instruments are rather similar, at least during the period where the humidity is above 200 ppmv (mid-December to end of January, Fig. 6c). The calibration of both analysers modifies the average  $\delta D$  values (mean difference of 7.4% for the Picarro and 19.1% for the AP2E analyser over the whole period, **Table 5**). The calibration of the  $\delta D$  time series does not affect the amplitude of the diurnal cycle for neither analyser (Fig. 7c). Both raw  $\delta D$  time series compare relatively well from mid-December to the end of January (dashed lines in Fig. 6c), with the same in-phase relationship between  $\delta D$  and the mixing ratio as for the calibrated  $\delta^{18}O$  time series. This in-phase relationship between  $\delta D$  and the humidity is preserved after calibration (plain lines in Fig. 6c and 7c).

Table 5. Root mean square difference (RMSD), mean difference (MD), mean absolute difference (MAD) and squared Pearson correlation coefficients (R2) between the raw and calibrated time series of the Picarro and AP2E analysers. MD is calculated as calibrated – raw. All statistics are calculated using the data between December 6th 2023 and March 15th 2024.

|                              | parameter               | RMSD      | MD        | MAD             | $\mathbb{R}^2$ |
|------------------------------|-------------------------|-----------|-----------|-----------------|----------------|
| Picarro raw vs calibrated    | humidity                | 26.8 ppmv | 23.6 ppmv | 23.6 ppmv       | 1.0            |
|                              | $\delta^{18}O$          | 17.8‰     | -9.3‰     | 9.3‰            | 0.49           |
|                              | δD                      | 13.2%     | 7.4%      | 11.8‰           | 0.98           |
|                              | d-excess                | 137.2%    | 81.9‰     | 81.9‰           | 0.53           |
| AP2E
raw vs
calibrated | humidity                | 2.6 ppmv  | 1.5 ppmv  | 2.3 ppmv | 1.0            |
|                              | $\delta^{18}\mathrm{O}$ | 10.5‰     | 9.2%      | 9.2‰            | 0.84           |
|                              | δD                      | 28.6‰     | 19.1‰     | 19.1‰           | 0.97           |
|                              | d-excess                | 58.2%     | -54.4‰    | 54.4%           | 0.77           |

**3.2.2 Instrument inter-comparison**

There is a good agreement between the  $\delta^{18}$ O calibrated time series from the AP2E and Picarro analysers over the period, until they start to diverge mid-February (Fig. 6b). Considering the entire period, the two  $\delta^{18}$ O calibrated time series show a mean absolute difference of 3.6% (**Table 6**), which is within the range of uncertainties of the calibrated time series (shaded areas in Fig. 6b), and a squared Pearson correlation coefficient R2 of 0.58 (**Table 6**). When considering only the period before mid-February, the mean absolute difference is reduced to 1.8% and R2 is improved to 0.8 (not shown). The good agreement between the two analysers confirms that the calibration is valid for the range of humidities encountered over this period.

As for  $\delta^{18}$ O, we observe that the calibrated  $\delta$ D time series from both instruments agree well between mid-December to mid-February, when similarly to  $\delta^{18}$ O they start to diverge (Fig. 6c). Over the entire period, the mean absolute difference between the two calibrated  $\delta$ D time series is 22.1% (**Table 6**), which is also within the uncertainty of both calibrated time series (shaded areas in Fig. 6c), and R2 is 0.59 (Fig. 6c). When considering only the period before mid-February, the mean absolute difference is reduced to 10.7% and R2 is improved to 0.85 (not shown).

Finally, the raw time series of d-excess are very different between the two analysers (Fig. 6d and 7d). However, after the calibration of both analysers, the two d-excess time series are comparable within their uncertainty range (Fig. 6d and 7d), with a mean absolute difference of 27.7‰ (Table 6). As for  $\delta^{18}$ O and  $\delta$ D, the calibrated d-excess time series of the two analysers diverge from mid-February onwards (Fig. 6d).

[...]

Table 6. Root mean square difference (RMSD), mean difference (MD), mean absolute difference (MAD) and squared Pearson correlation coefficients (R2) between the two calibrated time series of the Picarro and AP2E analysers. MD is calculated as Picarro – AP2E. All statistics are calculated using the data between December 6th 2023 and March 15th 2024.

| parameter               | RMSD     | MD        | MAD             | $\mathbb{R}^2$ |
|-------------------------|----------|-----------|-----------------|----------------|
| humidity                | 4.5 ppmv | -1.0 ppmv | 2.7 ppmv | 1.0            |
| $\delta^{18}\mathrm{O}$ | 6.7‰     | -3.5‰     | 3.6‰            | 0.58           |
| δD                      | 39.6‰    | -9.8‰     | 22.1‰           | 0.59           |
| d-excess                | 54.4%    | 18.6‰     | 27.7‰           | 0.02           |

Is it because of different temporal span or because of calibration that the statistics in grey in the first panel do not match with the statistics from Figure 2c?

It is both due to a different temporal span and the comparison between different datasets. The statistics in Figure 2c correspond to the comparison between the raw humidity time series measured by the Picarro and the AP2E analysers over the period January 1st to March 15th 2024. In Figure 6a, the statisctics correspond to the comparison between the calibrated humidity time series measured by the

Picarro and AP2E analysers over the period December 6th 2023 to March 15th 2024.

Figure 8: Same issue. Either make the statistics a separate table or show them with a scatter plot (potentially as additional panels on the right-hand side of the time series plots in Figure 8).

We removed the statistics from Figure 8 and included a new table in Section 3.3 (l. 503-530, bold indicates new/modified text):

[...]. Table 7 summarizes the statistics of the comparison between the modelled and calibrated time series from the Picarro and AP2E analysers.

The comparison of the humidity modelled by LMDZ6-iso and measured by both analysers shows an overall good agreement albeit a positive bias in the model (**Fig. 8a and Table 7**), including in terms of the amplitude of the observed diurnal cycle (Fig. 8a and 9a). However, during some specific periods, the model shows higher humidity levels than what is observed, especially during the nighttime (e.g. December 16th to 20th, light brown area in Fig. 8a). Contrary to the humidity, the air temperature modelled by LMDZ does not exhibit any mean bias compared to the air temperature measured by the local AWS (MD = -0.6°C and MAD = 2.8°C, not shown).

Although the model reproduces the observed in-phase relationship between  $\delta^{18}$ O and the humidity, the comparison between the modelled and observed  $\delta 18$ O shows a poorer agreement than for humidity. Firstly, the modelled  $\delta^{18}$ O shows an overall positive bias over the entire period compared to the observations, with a mean difference of 4.9% compared to the Picarro analyser and 3.0% compared to the AP2E analyser (**Table 7**). Secondly, the amplitude of the diurnal cycle modelled by LMDZ6-iso is overall larger than in the observations (Fig. 8b). Over the period January 11th to January 15th 2024 (Fig. 9b), the amplitude of the mean diurnal cycle in  $\delta^{18}$ O modelled by LMDZ6-iso is 10.9% (from -70.9 to -60.0%, not shown), higher than the one from both the Picarro analyser (5.7%, Sect. 3.2) and the AP2E analyser (4.7%, Sect. 3.2).

The same patterns are observed for  $\delta D$ . The modelled  $\delta D$  also shows an overall mean positive bias compared to the observations, with a mean difference of 27.3% compared to the Picarro analyser and 19.4% compared to the AP2E analyser (**Table 7**). The amplitude of the diurnal cycle is also larger in LMDZ6-iso than in the observations (Fig. 8c). Between January 11th and January 15th 2024 (Fig. 9c), the mean diurnal amplitude modelled by LMDZ6-iso is 69.0% (from -515.8 to -446.8%, not shown), which is higher than the observed one (34.9% for Picarro analyser, 29.5% for AP2E analyser, Sect. 3.2).

Lastly, due to the biases identified for  $\delta^{18}O$  and  $\delta D$ , the d-excess modelled by LMDZ6-iso also shows some discrepancies with the observations. The model shows an overall negative bias compared to the observations, with a mean difference over the whole period of 12.1% compared to the Picarro analyser and of 4.6% compared to the AP2E analyser (**Table 7**). The comparison of the amplitudes of the diurnal cycle is less conclusive than for  $\delta^{18}O$  and  $\delta D$ , due to the large uncertainties associated with the observations (Fig. 9d). However, we observe that the model still correctly captures the observed antiphase relationship between d-excess and  $\delta^{18}O$  (or  $\delta D$ ), with a maximum d-excess when  $\delta^{18}O$  is minimal, i.e. during the night, and a minimum d-excess when  $\delta^{18}O$  is maximal, i.e. during the day (Fig.

Table 7. Root mean square difference (RMSD), mean difference (MD), mean absolute difference (MAD) and squared Pearson correlation coefficients ( $R^2$ ) between the calibrated time series of the Picarro and AP2E analysers and the modelled time series by LMDZ6-iso. MD is calculated as model – observations. All statistics are calculated using the data between December 6th 2023 and February 14th 2024.

|                         | parameter           | RMSD      | MD        | MAD       | $\mathbb{R}^2$ |
|-------------------------|---------------------|-----------|-----------|-----------|----------------|
| LMDZ6-iso vs
Picarro | humidity            | 94.9 ppmv | 36.0 ppmv | 69.9 ppmv | 0.82           |
|                         | $\delta^{18}$ O     | 6.8‰      | 4.9‰      | 6.0%      | 0.45           |
|                         | $\delta { m D}$     | 40.1‰     | 27.3‰     | 35.0‰     | 0.49           |
|                         | d-excess            | 15.8‰     | -12.1‰    | 13.7‰     | 0.25           |
|                         | humidity            | 90.2 ppmv | 29.0 ppmv | 66.5 ppmv | 0.83           |
| LMDZ6-iso vs            | $\delta^{18}{ m O}$ | 6.4‰      | 3.0%      | 5.5‰      | 0.23           |
| AP2E                    | $\delta { m D}$     | 37.9‰     | 19.4‰     | 32.3‰     | 0.35           |
|                         | d-excess            | 21.0%     | -4.6%     | 17.0%     | 0.0            |